**1    Nationwide increase of polycyclic aromatic hydrocarbons in ultrafine particles during**

**2    winter over China revealed by size-segregated measurements**

Qingqing Yu[a], Xiang Ding[a,d,*], Quanfu He[a], Weiqiang Yang[e], Ming Zhu[a,b], Sheng Li[a,b], Runqi
Zhang[a,b], Ruqin Shen[a], Yanli Zhang[a,c,d], Xinhui Bi [a,d], Yuesi Wang[c,f], Ping'an Peng[a,d], Xinming
Wang[a,b,c,d,*]
[a]State Key Laboratory of Organic Geochemistry and Guangdong Key Laboratory of
Environmental Protection and Resources Utilization, Guangzhou Institute of Geochemistry,
Chinese Academy of Sciences, Guangzhou 510640, China
[b]University of Chinese Academy of Sciences, Beijing 100049, China
[c]Center for Excellence in Regional Atmospheric Environment, Institute of Urban Environment,
Chinese Academy of Sciences, Xiamen 361021, China
[d]Guangdong-Hong Kong-Macao Joint Laboratory for Environmental Pollution and Control,
Guangzhou Institute of Geochemistry, Chinese Academy of Science, Guangzhou 510640,
China
[e]Guangdong Provincial Academy of Environmental Science, Guangzhou 510045, China
[f]State Key Laboratory of Atmospheric Boundary Layer Physics and Atmospheric Chemistry,
Institute of Atmospheric Physics, Chinese Academy of Sciences, Beijing 100029, China
*corresponding author:
Dr. Xinming Wang and Dr. Xiang Ding
State Key Laboratory of Organic Geochemistry Guangzhou Institute of Geochemistry, Chinese
Academy of Sciences, 511 Kehua Rd, Tianhe, Guangzhou, 510640, China.
Email addresses: wangxm@gig.ac.cn and xiangd@gig.ac.cn

**Abstract**

Polycyclic aromatic hydrocarbons (PAHs) are toxic compounds in the atmosphere and have adverse effects on public health, especially through the inhalation of particulate matter (PM). At present, there are limited understandings in size distribution of particulate-bound PAHs and its health risk on a continental scale. In this study, we carried out simultaneously PM campaign from October, 2012 to September, 2013 at 12 sampling sites including urban, sub-urban and remote sites in different regions of China. Size-segregated PAHs and typical tracer of coal combustion (picene), biomass burning (levoglucosan) and vehicle exhaust (hopanes) were measured. The annual averages of total 24 PAHs ($\sum_{24}$PAHs) and benzo[a]pyrene (BaP) carcinogenic equivalent concentration (BaP$_{eq}$) ranged from 7.56 to 205 ng m$^{-3}$ with a mean of 53.5 ng m$^{-3}$ and 0.21 to 22.2 ng m$^{-3}$ with a mean of 5.02 ng m$^{-3}$, respectively. At all the sites, $\sum_{24}$PAHs and BaP$_{eq}$ were dominated in the ultrafine particles with aerodynamic diameter <1.1 μm, followed by those in the size ranges of 1.1-3.3 μm and >3.3 μm. Compared with the southern China, the northern China witnessed much higher $\sum_{24}$PAHs (87.36 ng m$^{-3}$ vs. 17.56 ng m$^{-3}$), BaP$_{eq}$ (8.48 ng m$^{-3}$ vs. 1.34 ng m$^{-3}$) and PAHs inhalation cancer risk (7.4×10$^{-4}$ vs. 1.2×10$^{-4}$). Nationwide increases in both PAH levels and inhalation cancer risk occurred in winter. The unfavorable meteorological conditions and enhanced emissions of coal combustion and biomass burning together led to severe PAHs pollution and high cancer risk in the atmosphere of the northern China, especially during winter. Coal combustion is the major source of BaP$_{eq}$ in all size particles at most sampling sites. Our results suggested that the reduction of coal and biofuel consumption in the residential sector could be crucial and effective to lower PAH concentrations and its inhalation cancer risk in China.

**Key words**: Polycyclic aromatic hydrocarbons; China; inhalation cancer risk; coal combustion;

biomass burning

## 1. Introduction

Ambient particulate matter (PM) pollution has adverse effects on public health. The global deaths caused by exposure to the PM with aerodynamic diameters less than 2.5 μm ($PM_{2.5}$) kept increasing from 1990 and reached 4.2 million in 2015 (Cohen et al., 2017). In China, ambient $PM_{2.5}$ pollution ranked the fourth leading risks for deaths (Yang et al., 2013), and caused 1.7 million premature deaths in 2015 (Song et al., 2017). Adverse health impacts of PM are associated with particle size and chemical components (Chung et al., 2015; Dong et al., 2018). Higher risk of cardiovascular disease was associated with smaller size-fractioned particulate matter, especially $PM_{1.0}$-bound particulate matter (Yin et al., 2020).

Polycyclic aromatic hydrocarbons (PAHs) are a group of organic substances composed of two or more aromatic rings. Due to the mutagenic, teratogenic, and carcinogenic properties (Kim et al., 2013), PAHs are one of the most toxic components in PM (Xu et al., 2008). Toxic PAHs usually enrich in fine particles, especially the aerodynamic diameters less than 1.0 μm (Wang et al., 2016; Li et al., 2019) which can enter the human respiratory system through inhalation (Yu et al., 2015). Exposure to PAHs likely induces DNA damage and raises the risk of gene mutation (Zhang et al., 2012; Lv et al., 2016) and cardiopulmonary mortality (Kuo et al., 2003; John et al., 2009). Previous studies have demonstrated that inhalation exposure to PAHs can cause high risk of lung cancer (Armstrong et al., 2004; Zhang et al., 2009; Shrivastava et al., 2017).

Atmospheric PAHs are mainly emitted from incomplete combustion of fossil fuels and biomasses (Mastral and Callen, 2000). As typical semi-volatile chemicals, PAHs can transport over long distances (Zelenyuk et al., 2012) and have been detected in the global atmosphere

(Brown et al., 2013; Garrido et al., 2014; Hong et al., 2016; Liu et al., 2017a; Hayakawa et al.,
2018). Emission inventory indicated that developing countries were the major contributors to
global PAHs emission (Zhang and Tao, 2009; Shen et al., 2013a).

As the largest developing country in the world, China has large amounts of PAHs emission

and high cancer risk caused by PAHs exposure. The annual emission of 16 USEPA priority
PAHs in China sharply increased from 18 Gg in 1980 to 106 Gg in 2007 (Xu et al., 2006; Shen
et al., 2013a). China became the largest emitter of PAHs, accounting for about 20% of the global
PAHs emission during 2007 (Shen et al., 2013a). The excess lung cancer risk caused by
inhalation exposure to ambient PAHs was estimated to be $6.5 \times 10^{-6}$ in China (Zhang et al., 2009),
which was 5.5 times higher than the acceptable risk level of $1.0 \times 10^{-6}$ in US (USEPA, 1991). As
Hong et al. (2016) estimated, the lifetime excess lung cancer cases caused by exposure to PAHs
for China ranged from 27.8-2200 per million people and were higher than other Asia counties.

Moreover, PAHs emission and cancer risk in China have large spatial and seasonal

variations. As reported by Tao and coworkers, high emission of PAHs occurred in the North
China Plain (Zhang et al., 2007), and the emission in winter was 1.6 times higher than that in
summer (Zhang and Tao, 2008). Thus, the lung cancer risk caused by ambient PAH inhalation
exposure in the northern China was higher than that in the southern China (Zhang et al. 2009).
In addition, through long-range atmospheric transport, PAHs emitted in China could spread to
other regions of the world (Zhang et al., 2011; Inomata et al., 2012).

For more accurate estimation of inhalation exposure to ambient PAHs and its cancer risks

in China, it is essential to carry out nationwide campaigns to acquire spatial and seasonal
characteristics of atmospheric PAHs. The data of PAHs in the ambient air are accumulating in
China during the past decades. Among these filed studies, most were conducted in rapidly
developing economic regions, including the North China region (Huang et al., 2006; Liu et al.,
2007a; Wang et al., 2011; Lin et al., 2015a; Lin et al., 2015b; Tang et al., 2017; Yu et al., 2018),
Yangtze River Delta region (Liu et al., 2001; Zhu et al., 2009; Gu et al., 2010; He et al., 2014)
and Pearl River Delta region (Bi et al., 2003; Guo et al., 2003; Li et al., 2006; Tan et al., 2006;
Duan et al., 2007; Lang et al., 2007; Yang et al., 2010; Gao et al., 2011, 2012, 2013, 2015; Yu
et al., 2016), due to large amounts of combustion emission and high density of population in
these regions. These studies provided insight into the fate and health risk of airborne PAHs on
a local or regional scale. However, due to the inconsistency in sampling methods, frequency
and duration in these local and regional campaigns, it is difficult to draw a national picture of
PAHs pollution in the air of China.

There are rare dataset discovering nationwide characteristics of airborne PAHs over China.

Liu et al. (2007b) reported PAHs in the air of 37 cities across China using passive polyurethane
foam (PUF) disks. Wang et al. (2006) and Liu et al., (2017b) determined $PM_{2.5}$-bound PAHs
over 14 and 9 Chinese cities, respectively. PAHs in the total suspended particle (TSP) and gas
phase were measured over 11 cities in China (Ma et al., 2018; Ma et al., 2020). Besides these
important information of PAHs in the bulk PM, it is vital to determine size distribution of PAHs,
since the size of particles is directly linked to their potential for causing health problems. On
the national scale, at present, there is only one field study available reporting size-segregated
atmospheric PAHs at 10 sites (Shen et al., 2019). Therefore, it is essential to carry out large
range campaigns coving multiple types of sites across different regions to investigate size
distribution of PAHs levels and sources and discover their difference in health risks among
typical regions of China (e.g. north vs. south, urban vs. remote). In this study, we
simultaneously collected filter-based size-fractionated PM samples consecutively at 12 sites for
one year. We analyzed chemical compositions of PAHs as well as other organic tracers to
characterize the spatiotemporal pattern and size distribution of PAHs over China and to explore
the possible sources of PAHs on the national scale. This information is helpful to provide a
basis for PAHs pollution control and health effects reduction in different regions of China.
**2. Materials and Methods**
2.1 Field sampling
The PM samples were collected simultaneously at 12 sampling sites across 6 regions of
China, containing five urban sites, three sub-urban sites and four remote sites (Figure S1 and
Table S1 in the supporting information). The Huai River-Qin Mountains Line is the
geographical line that divides China into the northern and southern regions. There are central
heating systems in winter in some urban areas of the northern China, but not so in the southern
China. The 12 sampling sites are Beijing (BJ), Dunhuang (DH), Hefei (HF), Hailun (HL),
Kunming (KM), Qianyanzhou (QYZ), Sanya (SY), Shapotou (SPT), Taiyuan (TY), Tongyu
(TYU), Wuxi (WX) and Xishuangbanna (BN). According to their locations, 6 of the 12 sites
are situated in the northern China, including BJ, DH, HL, SPT, TY and TYU. And the remaining
6 sites are located in the southern China, including BN, HF, KM, QYZ, SY and WX.
Total suspended particles (TSP) were collected using Anderson 9-stage cascade impactors
(<0.4, 0.4-0.7, 0.7-1.1, 1.1-2.1, 2.1-3.3, 3.3-4.7, 4.7-5.8, 5.8-9.0, >9.0 μm) at a constant flow of
28.3 L/min. Quartz fiber filters (Whatman, QMA) that were used to collect PM samples were
prebaked for 8 h at 450 ℃. At each site, one set of nine size-fractionated PM samples were
collected for 48-hr every 2 weeks. 294 sets of field samples and one set of field blanks were
collected. Detailed information of the field sampling can be found elsewhere (Ding et al., 2014).
According to the meteorological definition, each season lasts three months that spring runs from
March to May, summer runs from June to August, fall (autumn) runs from September to
November, and winter runs from December to February.

The data of average temperature (T), relative humidity (RH), the maximum solar radiation

(SR) during each sampling episode were available in the China Meteorological Data Service
Center (http://data.cma.cn/en). And the average boundary layer height (BLH) was calculated
using the NOAA's READY Archived Meteorology online calculating program
(http://ready.arl.noaa.gov/READYamet.php).
2.2 Chemical analysis

Each set of nine filters were combined into three samples with the aerodynamic diameters

smaller than 1.1 μm ($PM_{1.1}$), between 1.1 μm and 3.3 μm ($PM_{1.1-3.3}$), and large than 3.3 μm
($PM_{>3.3}$), respectively. Before ultrasonic solvent extraction, 400 ul of isotope-labeled mixture
compounds (tetracosane-$d_{50}$, napthalene-$d_8$, acenaphthene-$d_{10}$, phenethrene-$d_{10}$, chrysene-$d_{12}$,
perylene-$d_{12}$ and levoglucosan-$^{13}C_6$) were spiked into the samples as internal standards.
Samples were ultrasonic extracted twice with the mixed solvent of dichloride methane / hexane
(1:1, v/v), and then twice with the mixed solvent of dichloride methane / methanol (1:1, v/v).
The extracts of each sample were filtered, combined, and finally concentrated to about 1 mL.
Then the extracts were divided into two aliquots for silylation and methylation, respectively.
Detailed information about the procedures of silylation and methylation were introduced
elsewhere (Ding et al., 2014; Yu et al., 2016).
The methylated aliquot was analyzed for PAHs and hopanes using a 7890/5975C gas
chromatography/mass spectrometer detector (GC/MSD) in the selected ion monitoring (SIM)
mode with a 60 m HP-5MS capillary column (0.25 mm, 0.25 μm). The GC temperature was
initiated at 65 ℃, held for 2 min, and then increased to 300 ℃ at 5 ℃ min$^{-1}$ and held for 40
min. The silylated aliquot was analyzed for levoglucosan using the same GC/MSD in the scan
mode with a 30 m HP-5MS capillary column (0.25 mm, 0.25 μm). The GC temperature was
initiated at 65 ℃, held for 2 min, and then increased to 290 ℃ at 5 ℃ min$^{-1}$ and held for 20
min. The target compounds were identified by authentic standards and quantified using an
internal calibration approach. Table S2 lists the 24 target PAHs and their abbreviations.
2.3 Quality control and quality assurance
Field and laboratory blanks were analyzed in the same manner as the PM samples. The
target compounds were not detected or negligible in the blanks. The data reported in this study
were corrected by corresponding field blanks. To test the recovery of the analytical procedure,
we analyzed the NIST urban dust Standard Reference Material (SRM 1649b, n=6) in the same
manner as the PM samples. Compared with the certified values for PAHs in SRM 1649b, the
recoveries were 81.5±1.9%, 66.6±5.4%, 113.6±4.4%, 76.2±2.5%, 100.4±7.9%, 138.3±3.6%,
109.5±14.2%, 125.8±8.8% and 86.4±10.7% for Pyr, Ret, Chr, BbF, BkF, BeP, Per, IcdP and Pic
respectively. The data reported in this study were not recovery corrected. The method detection
limits (MDLs) of the target compounds ranged from 0.01 to 0.08 ng m$^{-3}$.
2.4 Positive matrix factorization (PMF) analysis
Positive matrix factorization (PMF) (USEPA, version PMF 5.0) was employed for source
apportionment of PAHs. The model has been widely used to attribute major sources of PAHs
(Larsen and Baker, 2003; Belis et al., 2011). In case the observed concentration (*Con*) of a
compound was below its MDL, half of the MDL was used as the model input data and the
uncertainty (*Unc*) was set as 5/6 of the MDL (Polissar et al., 1998). If the *Con* of a compound
was higher than its MDL, *Unc* was calculated as $Unc = [(20\% \times Con)^2 + (MDL)^2]^{1/2}$ (Polissar et
al., 1998).
2.5 Exposure assessment
Besides BaP, other PAHs like BaA, BbF, DahA and IcdP are also carcinogenic compounds
(IARC, 2001). In order to assess the carcinogenicity of bulk PAHs, the BaP carcinogenic
equivalent concentration ($BaP_{eq}$) was calculated by multiplying the concentrations of PAH
individuals ($PAH_i$) with their toxic equivalency factor ($TEF_i$) as:
$$BaP_{eq} = \sum_{i=1}^{n} PAH_i \times TEF_i \quad (1)$$
In this study, we adopted the TEFs reported by Nisbet and Lagoy (1992) which were 0.001
for Phe, Flu and Pyr, 0.01 for Ant, Chr and BghiP, 0.1 for BaA, BbF, BkF, BeP, and IcdP, and
1.0 for BaP and DahA. Table S3 lists annual averages of PAH individuals and $BaP_{eq}$ at the 12
sites.
Incremental lifetime lung cancer risk (ILCR) caused by inhalation exposure to PAHs was
estimated as:
$$ILCR = BaP_{eq} \times UR_{BaP} \quad (2)$$
where $UR_{BaP}$ is the unit relative risk of BaP. Based on the epidemiological data from studies in
coke-oven workers, the lung cancer risk of BaP inhalation was estimated to be $8.7 \times 10^{-5}$ per ng
$m^{-3}$ (WHO, 2000). Thus, we used a $UR_{BaP}$ value of $8.7 \times 10^{-5}$ per ng/m$^3$ in this study.
**3. Results and discussion**

**3.1 General marks**

Annual averages of the total 24 PAHs ($\sum_{24}$PAHs) in TSP (sum of three PM size ranges) ranged from 7.56 to 205 ng m$^{-3}$ (Figure 1a) among the 12 sampling sites with a mean of 53.5 ng m$^{-3}$. The highest concentration of $\sum_{24}$PAHs was observed at TY and the lowest level occurred at SY (Figure 1a). Compared with the data in other large scale observations (Table 1), atmospheric concentrations of PAHs measured at the 12 sites in this study were comparable with previously reported values in China in 2013-2014 (Liu et al., 2017b; Shen et al., 2019) and U.S. (Liu et al., 2017a), lower than those measured in China in 2003 and 2008-2009 (Wang et al., 2006; Ma et al., 2018), but higher than those over Great Lakes (Sun et al., 2006), Europe (Jaward et al., 2004), Japan (Hayakawa et al., 2018) and some Asian countries (Hong et al., 2016). Figure 1a also presents the compositions of PAHs. Apparently, 4- and 5-rings PAHs were the majority in $\sum_{24}$PAHs with the mass shares of 36.8±5.6% and 31.4±9.6%, respectively, followed by the PAHs with 3-rings (19.2±9.4%), 6-rings (11.3±3.8%), and 7-rings (1.3±0.6%). The concentrations of $\sum_{24}$PAHs at urban sites (82.7 ng m$^{-3}$) were significant higher (p<0.05) than those at sub-urban (48.0 ng m$^{-3}$) and remote sites (18.0 ng m$^{-3}$) (Figure S2).

Annual averages of BaP in TSP among the 12 sites were in the range of 0.09 to 11.0 ng m$^{-3}$ with a mean of 2.58 ng m$^{-3}$. The highest level of atmospheric BaP occurred at TY and the lowest existed at SY. The BaP values at five sites (WX, BJ, HL, DH and TY) exceeded the national standard of annual atmospheric BaP (1.0 ng m$^{-1}$) by factors of 1.2 to 11.0. For BaP$_{eq}$, annual averages ranged from 0.21 to 22.2 ng m$^{-3}$ with the predominant contribution from 5-rings PAHs (Figure 1b). ILCR caused by inhalation exposure to PAHs ranged from $1.8 \times 10^{-5}$ (SY) -$1.9 \times 10^{-3}$ (TY) among the 12 sites in China (Figure S3), which were much higher than the

acceptable risk level of $1.0 \times 10^{-6}$ in US (USEPA, 1991). All these demonstrated that China faced
severe PAHs pollution and high health risk (Zhang et al., 2009; Shrivastava et al., 2017). And
$BeP_{eq}$ (Figure S4) and ILCR (Figure S5) were both the highest at urban sites. All these indicated
that people in urban regions of China were faced with higher exposure risk of PAHs pollution
as compared to those in sub-urban and remote areas. Figure S6 exhibits that 4- and 5-rings
PAHs are the majority in $\sum_{24}$PAHs at urban, sub-urban and remote sites, which totally accounted
72.2%, 63.8% and 66.6% of the total amounts in TSP, respectively. The percentage of 5-rings
PAHs dominates at urban sites, and 4-rings PAHs makes the largest proportion at sub-urban and
remote sites (Figure S6).
**3.2 Enrichment of PAHs in PM$_{1.1}$**
Figure 2 presents the size distribution of PAHs and $BaP_{eq}$ at the 12 sites in China. Both
$\sum_{24}$PAHs and $BaP_{eq}$ were concentrated in PM$_{1.1}$, accounting for 44.6-71.3% and 56.7-79.3% of
the total amounts in TSP, respectively. And $BaP_{eq}$ had more enrichment in PM$_{1.1}$ than $\sum_{24}$PAHs.
The mass fractions of $\sum_{24}$PAHs and $BaP_{eq}$ in PM$_{1.1-3.3}$ were 20.6-39.5% and 16.1-38.3%. The
coarse particles (PM$_{>3.3}$) had the lowest loadings of $\sum_{24}$PAHs (7.2-23.4%) and $BaP_{eq}$ (3.0-
12.9%). Thus, our observations indicated that PAHs in the ultrafine particles (PM$_{1.1}$) contributed
most health risk of PAHs in TSP over China. A previous study at three sites in East Asia found
that size distribution of PAHs was unimodal and peaked at 0.7-1.1 μm size (Wang et al., 2009).
A recent study at 10 sites of China also found that PAHs were concentrated in PM$_{1.1}$ (Shen et
al., 2019). Based on the observation at one site in the Fenhe Plain, northern China, Li et al.
(2019) pointed out that PAHs in the particles with the aerodynamic diameters <0.95 μm
contributed more than 60% to the total cancer risk of PAHs in PM$_{10}$. All these results
demonstrate that high carcinogenicity of PAHs is accompanied with ultrafine particles,
probably because small particles are apt to invade the blood vessels and cause DNA damage.
Thus, further studies should put more attentions on PAHs pollution in ultrafine particles.

Figure S7 and Figure S8 show seasonal variations in size distribution of $\sum_{24}$PAHs and

BaP$_{eq}$, respectively. $\sum_{24}$PAHs and BaP$_{eq}$ were enriched in PM$_{1.1}$ throughout the year at all sites.
The mass fractions of $\sum_{24}$PAHs and BaP$_{eq}$ in PM$_{1.1}$ were the highest during fall to winter (up to
74.6% and 79.7% at the DH site), and the lowest during summer (down to 39.2% and 50.7% at
the BN site). It should be related to the emission sources of PAHs. Atmospheric PAHs are
mainly derived from combustion sources. As Shen et al. (2013b) reported, PAHs emitted form
biomass burning and coal combustion enriched in ultrafine particles (<1.1 μm). Moreover, coal
combustion witnessed more enrichment of PAHs in ultrafine particles than biomass burning.
Figure S9 presents monthly variations in size distribution of PAHs with different number of
rings. The mass shares of 3-rings PAHs in PM$_{1.1}$ (39.2%), PM$_{1.1-3.3}$ (32.0%) and PM$_{>3.3}$ (28.9%)
were comparable. And the highest loading of 3-rings PAHs in PM$_{1.1}$ was observed in December
2012. The mass fractions of 4-ring PAHs in PM$_{1.1}$ were the highest in December 2012 (58.4%)
and the lowest in July 2013 (39.5%). The higher molecular weight PAHs (5-7 rings PAHs) were
enriched in PM$_{1.1}$ throughout the year.
**3.3 High levels of atmospheric PAHs in the northern China**

Figure 3 shows the differences of atmospheric PAHs between the northern China (BJ, DH,

HL, SPT, TY and TYU) and southern China (BN, HF, KM, QYZ, SY and WX). $\sum_{24}$PAHs in
the northern China was higher than that in the southern China by a factor of 5.0 (Figure 3a).
The concentrations of PAHs with different ring number were all higher in the northern China
than those in the southern China, especially for the 4-7 rings PAHs. Moreover, BaP, $BaP_{eq}$ and
ILCR in the northern China were 5.8, 5.3 and 5.3 times higher than those in the southern China
(Figure 3b). The higher concentrations of PAHs in the air of the northern China than the
southern China were also reported in previous field studies (Liu et al., 2017b; Ma et al., 2018;
Shen et al., 2019). Based on the emission inventories and model results, previous studies
predicted that PAHs concentrations, BaP levels and lung cancer risk of exposure to ambient
PAHs in the northern China were all higher than those in the southern China (Xu et al., 2006;
Zhang et al., 2007; Zhang and Tao, 2009; Zhu et al., 2015). All these indicated much higher
PAHs pollution and health risk in the northern China.

The northern-high feature of atmospheric PAHs should be determined by the

meteorological conditions and source emissions. Theoretical relationship between
meteorological parameters (temperature, solar radiation and boundary layer height) and the
concentration of particulate-bound PAHs were discussed, the detail theoretical discussion
information can be found in Text S1 in the supporting information. We illustrate that decrease
of ambient temperature would result in the increase of individual PAH in the particulate phase
assuming a constant total concentration in the air. The decrease of SR can indeed lower
concentrations of hydroxyl radical [OH] and accumulate PAHs in the air, resulting in the
increase of PAHs concentrations. And low height of boundary layer can inhibit the vertical
diffusion of PAHs, which leads to PAHs accumulation and increased concentrations. As Figure
4 showed, PAHs exhibited strong negative correlations with temperature (T), solar radiation
(SR) and the boundary layer height (BLH), especially in the northern China. This indicated that
the unfavorable meteorological conditions, such as low levels of temperature, solar radiation
and BLH could lead to PAHs accumulation in the air (Sofuoglu et al., 2001; Callén et al., 2014;
Lin et al., 2015a; Li et al., 2016a). In fact, annual averages of T, SR and BLH in the northern
China were all significant lower than those in the southern China (p<0.05, Table S4), which
could indeed cause the accumulation of PAHs in the air of the northern China. In addition, low
temperature in the northern China would promote the condensation of semi-volatile PAHs on
particles (Wang et al., 2011; Ma et al., 2020). At the southern sites, the negative correlations
between PAHs and meteorological parameters (SR and BLH) were not as strong as those in the
northern sites. This implied that the adverse influence of meteorological conditions on PAHs
pollution in the southern China might be less significant than that in the northern China. The
annual ambient temperature at the 12 sampling sites was 13.9 ℃, then we choose 13.9 ℃ to
divide the one-year data into warm and cold seasons. As Figure S10 showed, at most sites in
the northern and southern China, PAHs negatively correlated with temperature (T), boundary
layer height (BLH) and solar radiation (SR) in both cold (T < 13.9 ° C) and warm (T > 13.9 °
C) seasons. Thus, coupled with theoretical discussion, we suggested that worsened PAH
pollution in winter was partly caused by adverse meteorological conditions.

For PAHs emission, there are apparent differences in sources and strength between the

northern and southern regions. For instance, there is central heating during winter in the
northern China, but not so in the southern China. The residential heating during cold period in
the northern China could consume large amounts of coal and biofuel, and release substantial
PAHs into the air (Liu et al., 2008; Xue et al., 2016). Consequently, atmospheric levels of PAHs
in the northern China were much higher than those in the southern China. Since central heating
systems start heat supply simultaneously within each region in the northern China, atmospheric
PAHs should increase synchronously within the northern regions of China. To check the spatial
homogeneity of PAHs on a regional scale, we analyzed the correlation of PAHs between paired
sites within each region. As Table 2 exhibited, PAHs varied synchronously and correlated well
at the paired sites in the northern China (p<0.001). And closer distance between sites, stronger
correlations were observed. The spatial synchronized trends of PAHs observed in the northern
regions of China probably resulted from the synchronous variation of PAHs emission in the
northern China. In the southern China, although the distances between paired sites were closer
than those in the northern regions, the correlations between sites within a region was weaker.
This indicated that there might be more local emission which sources and strength vary place
to place in the southern China.

We applied diagnostic ratios of PAH isomers to identify major sources of atmospheric

PAHs. The ratios of IcdP/(IcdP+BghiP) and Flu/(Flu+Pyr) have been widely used to distinguish
possible sources of PAHs (Aceves and Grimalt, 1993; Zhang et al., 2005; Ding et al., 2007;
Gao et al., 2012; Lin et al., 2015a; Ma et al., 2018). As summarized by Yunker et al. (2002), the
petroleum boundary ratios for IcdP/(IcdP+BghiP) and Flu/(Flu+Pyr) are close to 0.20 and 0.40,
respectively; for petroleum combustion, the ratios of IcdP/(IcdP+BghiP) and Flu/(Flu+Pyr)
range from 0.20 to 0.50 and 0.40 to 0.50, respectively; and the combustions of grass, wood and
coal have the ratios higher than 0.50 for both IcdP/(IcdP+BghiP) and Flu/(Flu+Pyr). As Figure
5 showed, the ratios of Flu/(Flu+Pyr) at the 12 sites ranged from 0.49 to 0.76, suggesting that
biomass (grass/wood) burning and coal combustion were the major sources. And the ratios of
IcdP/(IcdP+BghiP) were in the range of 0.32 to 0.62, indicating that besides biomass and coal
combustion, petroleum combustion, especially vehicle exhaust was also an important source of

atmospheric PAHs. Thus, as identified by the diagnostic ratios, biomass burning, coal

combustion and petroleum combustion were major sources of atmospheric PAHs over China.

This is also confirmed by the significant correlations of $\sum_{24}$ PAHs with the typical tracers of

biomass burning (levoglucosan), coal combustion (picene) and vehicle exhaust (hopanes) at

most sites (Figure 6). As global emission inventories showed, PAHs in the atmosphere were

mainly released from the incomplete combustion processes including coal combustion, biomass

burning and vehicle exhaust (Shen et al., 2013a).

To further attribute PAHs sources, we employed the PMF model to quantify source

contributions to atmospheric PAHs at the 12 sites in China. Three factors were identified, and

the factor profile resolved by PMF were presented in Figure S11. The first factor was identified

as biomass burning, as it had high loadings of the biomass burning tracer, levoglucosan and

light weight molecular PAHs such as Phe, Ant, Flu and Pyr which are largely emitted from

biomass burning (Li et al., 2016b). The second factor was considered to be coal combustion, as

it was characterized by high fractions of the coal combustion marker, picene and the high

molecular weight PAHs (Shen et al., 2013b). The third factor was regarded as vehicle exhaust,

as it was featured by presence of hopanes, which are molecular markers tracking vehicle

exhaust (Cass, 1998; Dai et al., 2015). As Figure S12 showed, there was significant agreement

between the predicted and measured PAHs at each site ($R^2$ in the range of 0.78 to 0.99, p<0.001).

As the emission inventory of PAHs in China showed, residential/commercial, industrial and

transportation were the major sectors of atmospheric PAHs in 2013 (Figure S13,

http://inventory.pku.edu.cn). Residential/commercial and industrial sectors mainly consumed

coal and biofuel while transportation consumed oil (Shen et al., 2013a). Thus, the mainly

sources of PAHs in China were coal combustion, biomass burning and petroleum combustion
(especially vehicle exhaust).
Figure 7a presents atmospheric PAHs emitted from different sources in China. In the
northern China, coal combustion was the major source of atmospheric PAHs (73.6 ng m$^{-3}$, 84.2%
of $\sum_{24}$PAHs), followed by biomass burning (11.8 ng m$^{-3}$ and 13.5%) and vehicle exhaust (2.0
ng m$^{-3}$ and 2.3%). In the southern China, coal combustion (9.6 ng m$^{-3}$ and 54.8%) and biomass
burning (6.8 ng m$^{-3}$ and 39.0%) were the major contributors, followed by vehicle exhaust (1.1
ng m$^{-3}$ and 6.2%). Atmospheric PAHs emitted from the three sources in the northern China were
all higher than those in the southern China, especially from coal combustion. Thus, coal
combustion was the most important source of atmospheric PAHs in China and caused large
increases in PAHs pollution in the northern China. As China statistics yearbook recorded
(http://www.stats.gov.cn/english/Statisticaldata/AnnualData/), coal was the dominant fuel in
China, accounting for 70.6% (24.1 $\times 10^8$ tons of Standard Coal Equivalent, SCE) of total primary
energy consumption in 2012, followed by crude oil 19.9% (6.7 $\times 10^8$ tons of SCE) and other
types of energy 9.5%, including biofuel, natural gas, hydro power, nuclear power and other
power (3.2 $\times 10^8$ tons of SCE). Although the biofuel consumption was lower than crude oil, the
poor combustion conditions during residential biofuel burning could led to higher PAHs
emissions as compared to petroleum combustion.
We further compared our results with those in the PAHs emission inventory of China
(http://inventory.pku.edu.cn) (Figure S14). Our source apportionment results focused on fuel
types, while the emission inventory classified the sources into 6 socioeconomic sectors
(residential & commercial activities, industry, energy production, agriculture, deforestation &
wildfire, and transportation). Since the transportation mainly used liquid petroleum (gasoline
and diesel) and the rest sectors mainly consumed solid fuels (coal and biomass), we grouped
these sectors into liquid petroleum combustion and solid fuel burning to directly compare with
our results. As Figure S14 showed, both our observation and emissions inventory demonstrated
that the PAHs contributions from solid fuel burning was higher in the northern China, while the
PAHs contributions from liquid petroleum combustion was higher in the southern China.

Atmospheric PAHs emitted from different sources at urban, sub-urban and remote sites

(Figure 7b) and different size particles (Figure 7c) were discussed. At urban and sub-urban sites,
coal combustion was the largest source of $\sum_{24}$PAHs (70.4 ng m$^{-3}$, 85.1% and 30.5 ng m$^{-3}$, 63.5%),
followed by biomass burning (10.1 ng m$^{-3}$, 12.2% and 16.3 ng m$^{-3}$, 33.9%) and vehicle emission
(2.2 ng m$^{-3}$, 2.6% and 1.2 ng m$^{-3}$, 2.5%), while at remote sites the contributions of coal
combustion (9.1 ng m$^{-3}$, 50.6% ) and biomass burning (7.8 ng m$^{-3}$, 43.7%) were comparable
and vehicle emission (1.0 ng m$^{-3}$, 5.7%) had minor contributions. The major sources of
$\sum_{24}$PAHs varied among different size particles in the northern and southern China (Figure 7c).
For PM$_{>3.3}$-bound PAHs, the contributions of coal combustion (50.3%) and biomass burning
(48.4%) were comparable in the northern China, while biomass burning (71.0%) was the largest
source in the southern China. For PM$_{1.1-3.3}$-bound PAHs, coal combustion (66.7%) was the
dominated source in the northern China, whereas the percentage of biomass burning (53.7%)
was larger than that of coal combustion (40.4%) in the southern China. For PM$_{1.1}$-bound PAHs,
coal combustion was the dominated source in the northern (66.6%) and southern (59.3%) China.

Source apportionment of BaP$_{eq}$ in different regions (Figure 7d), sampling sites (Figure 7e)

and size particles (Figure 7f) were also discussed. Unlike $\sum_{24}$PAHs, coal combustion was the
predominant source of $BaP_{eq}$ in the northern (8.1 ng m$^{-3}$ and 95.7%) and the southern (1.1 ng
m$^{-3}$ and 84.7%) China. The contributions of coal contribution at urban sites (8.3 ng m$^{-3}$ and
96.4%) were larger than those at sub-urban (3.3 ng m$^{-3}$ and 90.8%) and remote (1.0 ng m$^{-3}$ and
82.5%) sites. Coal combustion was the dominating source in different size particles. And its
contributions to $PM_{>3.3}$, $PM_{1.1-3.3}$ and $PM_{1.1}$-bound PAHs in the northern China (87.3%, 95.6%
and 96.9%) were all larger than those in the southern China (76.8%, 87.3% and 88.2%).
In terms of incremental lifetime lung cancer risk (ILCR) induced by ambient PAHs, coal
combustion was the largest source to total ILCR, accounting for 95.7% ($7.1 \times 10^{-4}$) and 84.7%
($1.0 \times 10^{-4}$) in the northern and southern China, respectively (Figure S15). The ILCR due to coal
combustion was as high as $1.9 \times 10^{-3}$ at the TY site in Shanxi province, which was three orders
of magnitude higher than the acceptable risk level of $1.0 \times 10^{-6}$ recommended by USEPA (1991).
Shanxi province has the largest coal industry in China, including coal mining and coking
production. Previous studies have reported that higher lung cancer risks occurred in Shanxi
province, largely owing to the extremely high inhalation exposure of PAHs there (Xia et al.,
2013; Liu et al., 2017b; Han et al., 2020). It should be noted that although the contributions of
biomass burning (2.1%, $1.6 \times 10^{-5}$ vs. 6.4%, $7.5 \times 10^{-6}$) and vehicle emission (2.2%, $1.6 \times 10^{-5}$ vs.
8.9%, $1.0 \times 10^{-5}$) to total ILCR were minor in the northern and southern China, their ILCR were
both exceed the acceptable risk level of $1.0 \times 10^{-6}$ (USEPA, 1991). Thus, the health risks from
biomass burning and vehicle emission cannot be ignored.
Figure S16 shows different source contributions to ILCR at the urban, sub-urban and
remote sites. Coal combustion was the dominant source to total ILCR, which accounted for
96.4% ($7.2 \times 10^{-4}$) at the urban sites, 90.8% ($2.9 \times 10^{-4}$) at the sub-urban sites, and 82.5% ($8.6 \times 10^{-}$
$^5$) at the remote sites. The ILCR from biomass burning were the highest at the urban sites
($1.3\times10^{-5}$), followed by the sub-urban ($1.2\times10^{-5}$) and remote sites ($9.5\times10^{-6}$). For vehicle
emission, the ILCR were $1.4\times10^{-5}$, $1.7\times10^{-5}$, and $8.7\times10^{-6}$ at the urban, sub-urban and remote
sites. Our results indicated that even the remote areas in China would face high health risks
since the ILCR from the least contributor (e.g. $8.7\times10^{-6}$ for vehicle emission) were exceed the
acceptable risk level of $1.0\times10^{-6}$ (USEPA, 1991).

Here, we concluded that the unfavorable meteorological conditions and intensive emission

especially in coal combustion together led to severe PAHs pollution and high cancer risk in the
atmosphere of the northern China.
**3.4 Nationwide increase of PAHs pollution and health risk during winter**

Figure 8 exhibits monthly variations of $BaP_{eq}$ and ILCR at the 12 sites. $BaP_{eq}$ levels were

the highest in winter and the lowest in summer at all sites. As Figure 8 showed, the enhancement
of $BaP_{eq}$ from summer to winter ranged from 1.05 (SY) to 32.5 (SPT). And such an
enhancement was much more significant at the northern sites than the southern sites. Hence,
ILCR was significantly enhanced in winter, especially in the northern China (Figure 8) and was
much higher than the acceptable risk level of $1.0\times10^{-6}$ in US (USEPA, 1991). Previous studies
in different cities of China also reported such a winter-high trend of atmospheric PAHs (Liu et
al., 2017b; Ma et al., 2018; Shen et al., 2019). Thus, there is a nationwide increase of PAHs
pollution during winter in China.

The winter-high feature of PAHs pollution should result from the impacts of

meteorological conditions and source emissions. The winter to summer ratios of PAHs
correlated well with that for temperature (Figure S17). And T, SR and BLH were all the lowest
during winter and the highest during summer (Table S5-7). Coupled with the negative
correlations between PAHs and meteorological factors (Figure 4), the unfavorable
meteorological conditions in wintertime did account for the increase in PAHs pollution.

Moreover, PAHs emitted from coal combustion and biomass burning apparently elevated

during fall-winter (Figure 9). In the northern China, central heating systems in urban areas
usually start from November to next March. Meanwhile residential heating in the rural areas of
northern China consumes substantial coal and biofuel (Xue et al., 2016). Thus, the energy
consumption in the residential sector is dramatically enhanced during fall-winter (Xue et al.,
2016). In the southern China, although there is no central heating system in urban areas, power
plant and industry consume large amounts of coal. And there is also residential coal/biofuel
consumption for heating during winter as well as cooking in rural areas (Zhang et al., 2013; Xu
et al., 2015). In addition, open burning of agriculture residuals which accounts for a major
fraction of the total biomass burning in China will significantly increase during fall-winter
harvest seasons in the southern China (Zhang et al., 2013). Our observation and emissions
inventory witnessed similar monthly trends that the PAHs from solid fuel combustion (coal and
biomass) apparently elevated during fall-winter in the northern and southern China (Figure S18).
Previous field studies also found that the contributions of coal combustion and biomass burning
to PAHs elevated during fall-winter (Lin et al., 2015a; Yu et al., 2016). Thus, we concluded that
the unfavorable meteorological conditions and intensive source emission together led to the
increase of PAHs pollution during winter.

Figure S19 presents seasonal variation of ILCR from different sources. The ILCR values

from three major sources all elevated during winter. Coal combustion was the largest source to
ILCR, accounting for 94.4% ($4.2 \times 10^{-4}$), 94.1% ($10.8 \times 10^{-4}$), 89.2% ($1.8 \times 10^{-4}$) and 83.8%
($6.5 \times 10^{-5}$) in fall, winter, spring and summer, respectively. The ILCR from biomass burning
was highest in winter ($3.7 \times 10^{-5}$), followed by spring ($1.1 \times 10^{-5}$), fall ($9.1 \times 10^{-6}$) and summer
($7.9 \times 10^{-6}$). For vehicle emission, the ILCR were $1.6 \times 10^{-5}$, $3.0 \times 10^{-5}$, $1.1 \times 10^{-5}$ and $4.7 \times 10^{-6}$ in
fall, winter, spring and summer, respectively. Our results revealed that even in summer people
would face high health risks since the ILCR from the least contributor (e.g. $4.7 \times 10^{-6}$ for vehicle
emission) was exceed the acceptable risk level of $1.0 \times 10^{-6}$ (USEPA, 1991).
**Data availability**
The data are given in the Supplement.
**Author contributions**
Qingqing Yu analyzed the data, wrote the paper and performed data interpretation. Quanfu He
and Ruqin Shen analyzed the samples. Weiqiang Yang ran the PMF model and helped with the
interpretation. Ming Zhu, Sheng Li and Runqi Zhang provided the meteorological data and
prepared the related interpretation. Yanli Zhang and Xinhui Bi gave many suggestions about
the results and discussion. Yuesi Wang helped the field observation and performed data
interpretation. Xiang Ding, Ping'an Peng and Xinming Wang performed data interpretation,
reviewed and edited this paper.
**Competing interests**
The authors declare that they have no conflict of interest.
**Acknowledgement**
This study was funded by the National Natural Science Foundation of China
(41530641/4191101024/41722305/41907196), the National Key Research and Development
Program (2016YFC0202204/2018YFC0213902), the Chinese Academy of Sciences
(XDA05100104/QYZDJ-SSW-DQC032), and Guangdong Foundation for Science and
Technology Research (2019B121205006/2017BT01Z134/ 2020B1212060053).

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

Table 1 PAHs concentration measured in this study and comparison with those of other large scale
observations.

| Site/type | Sampling period | Sample type | # of sites | # of species | PAHs (ng/m$^3$) | Reference |
|-----------|-----------------|-------------|------------|--------------|-----------------|-----------|
| China[a] | Oct, 2012-Sep, 2013 | $PM_{1.1}$ | 12 | 24 | 3.4-126.2 | This study |
| China[a] | Oct, 2012-Sep, 2013 | $PM_{1.1-3.3}$ | 12 | 24 | 2.4-55.7 | This study |
| China[a] | Oct, 2012-Sep, 2013 | $PM_{>3.3}$ | 12 | 24 | 1.8-22.7 | This study |
| China/Urban | 2003 | $PM_{2.5}$ | 14 | 18 | 1.7-701 | Wang et al., 2006 |
| China[b] | 2005 | PUF | 40 | 20 | 374.5[e] | Liu et al., 2007 |
| China/Urban | 2013-2014 | $PM_{2.5}$ | 9 | 16 | 14-210 | Liu et al., 2017b |
| China/Urban | Aug, 2008-July, 2009 | $PM_{2.5}$ | 11 | 16 | 75.4-478 | Ma et al., 2018 |
| China[c] | Jan, 2013-Dec, 2014 | $PM_{9.0}$[e] | 10 | 12 | 17.3-244.3 | Shen et al., 2019 |
| Great Lakes | 1996-2003 | PUF | 7 | 16 | 0.59-70 | Sun et al., 2006 |
| Asian countries[d] | Sep, 2012-Aug, 2013 | PUF | 176 | 47 | 6.29-688 | Hong et al., 2016 |
| U.S. | 1990-2014 | PUF | 169 | 15 | 52.6 | Liu et al., 2017a |
| Japan | 1997-2014 | TSP | 5 | 9 | 0.21-3.73 | Hayakawa et al., 2018 |
| Europe | 2002 | PUF | 22 | 12 | 0.5-61.2 | Jaward et al., 2004 |

a: including 5 urban sites, 3 sub-urban sites and 4 remote sites in China
b: including 37 cities and 3 rural locations in China
c: including 5 urban sites, 1 sub-urban site, 1 farmland site and 3 background sites in China
d: including 82 urban sites, 83 rural sites and 11 background sites in China, Japan, South Korea,
Vietnam, and India
e: the unit was ng/day

Table 2 Correlation coefficient (r), significance (p) of PAHs between paired sites in each region.

| regions | Northern China | | | Southern China | |
|---|---|---|---|---|---|
| | north | northeast | northwest | east | southwest |
| paired sites | BJ-TY | HL-TYU | DH-SPT | WX-HF | KM-BN |
| distance between sites | 400 km | 450 km | 940 km | 280 km | 380 km |
| r | 0.97 | 0.80 | 0.63 | 0.77 | - |
| p | <0.001 | <0.001 | 0.001 | <0.001 | 0.09 |




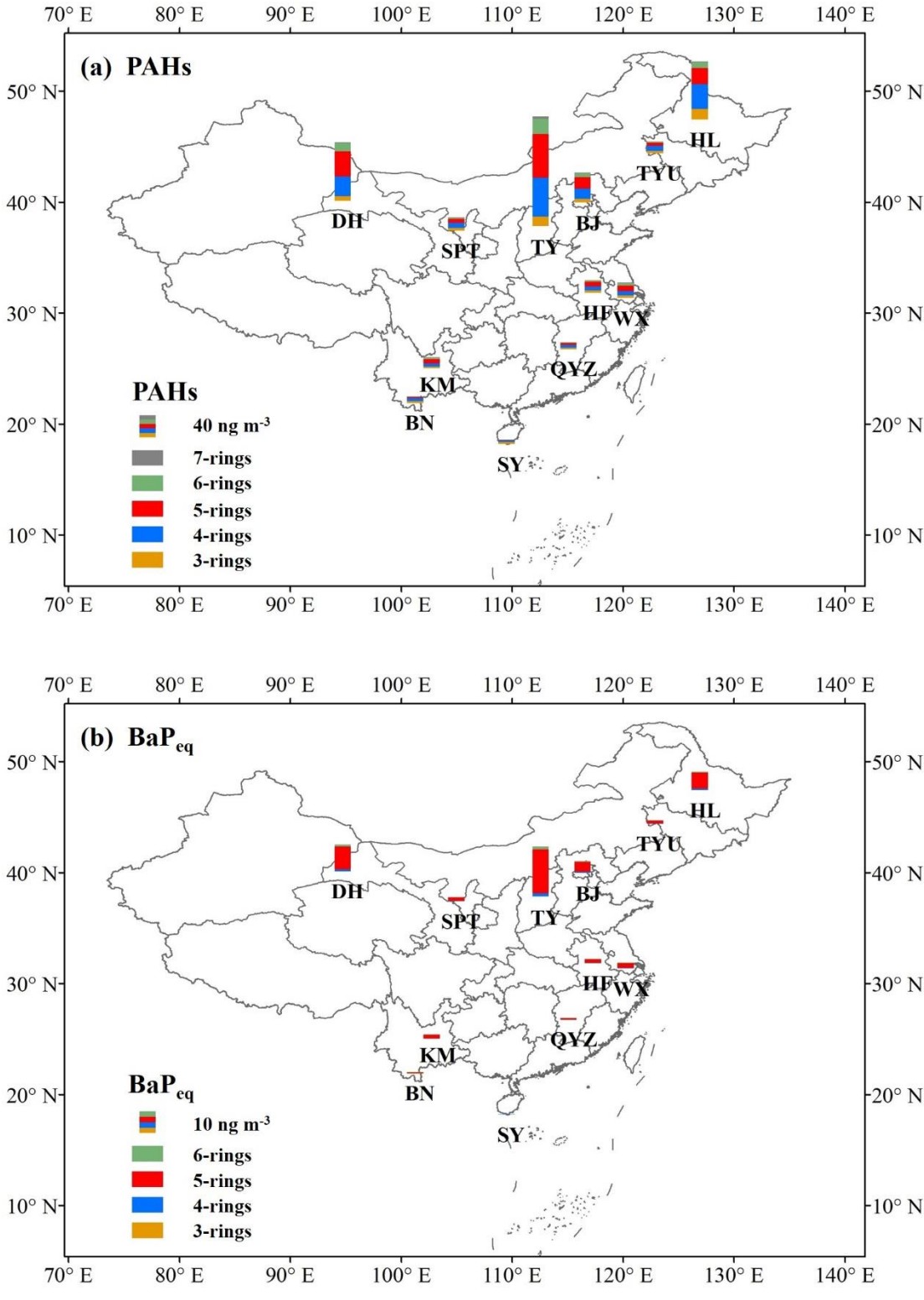


Figure 1 Annual averages of $\sum_{24}$PAHs (a) and BaP$_{eq}$ (b) at 12 sites in China.

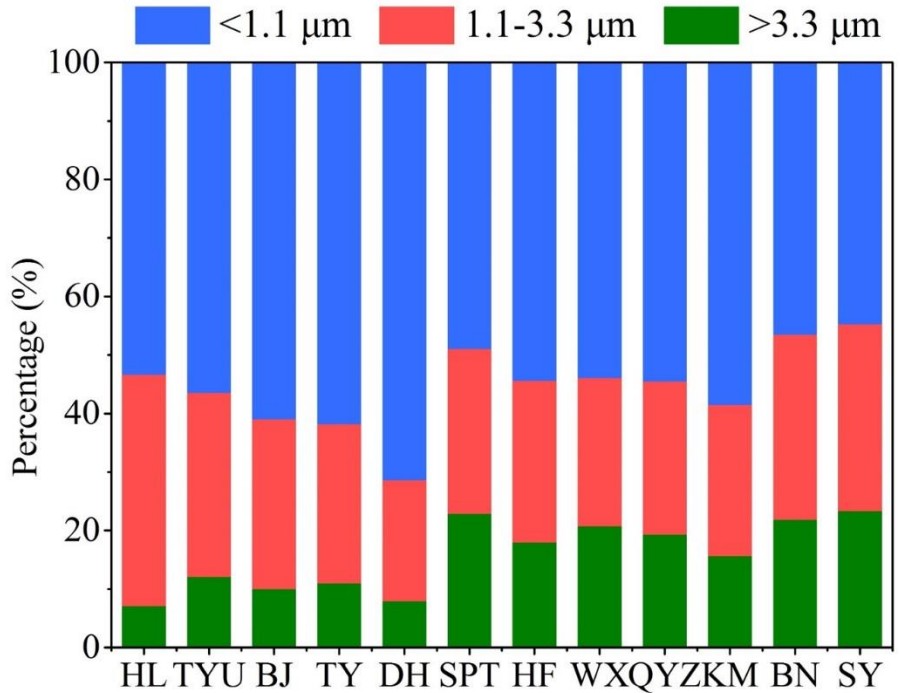

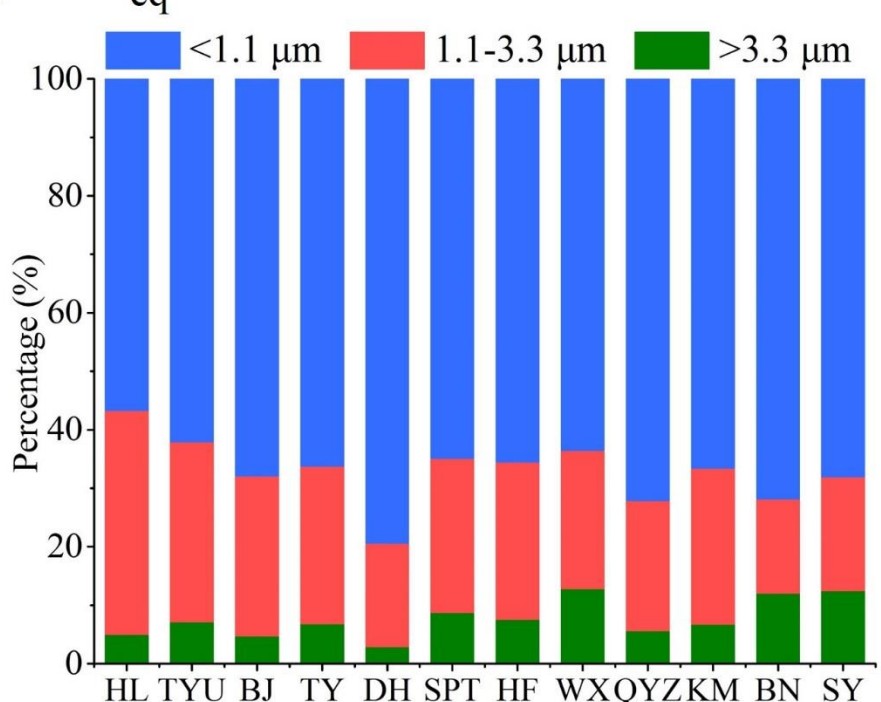


Figure 2 Size distribution of total measured PAHs (a) and BaP$_{eq}$ (b) at 12 sites over China.

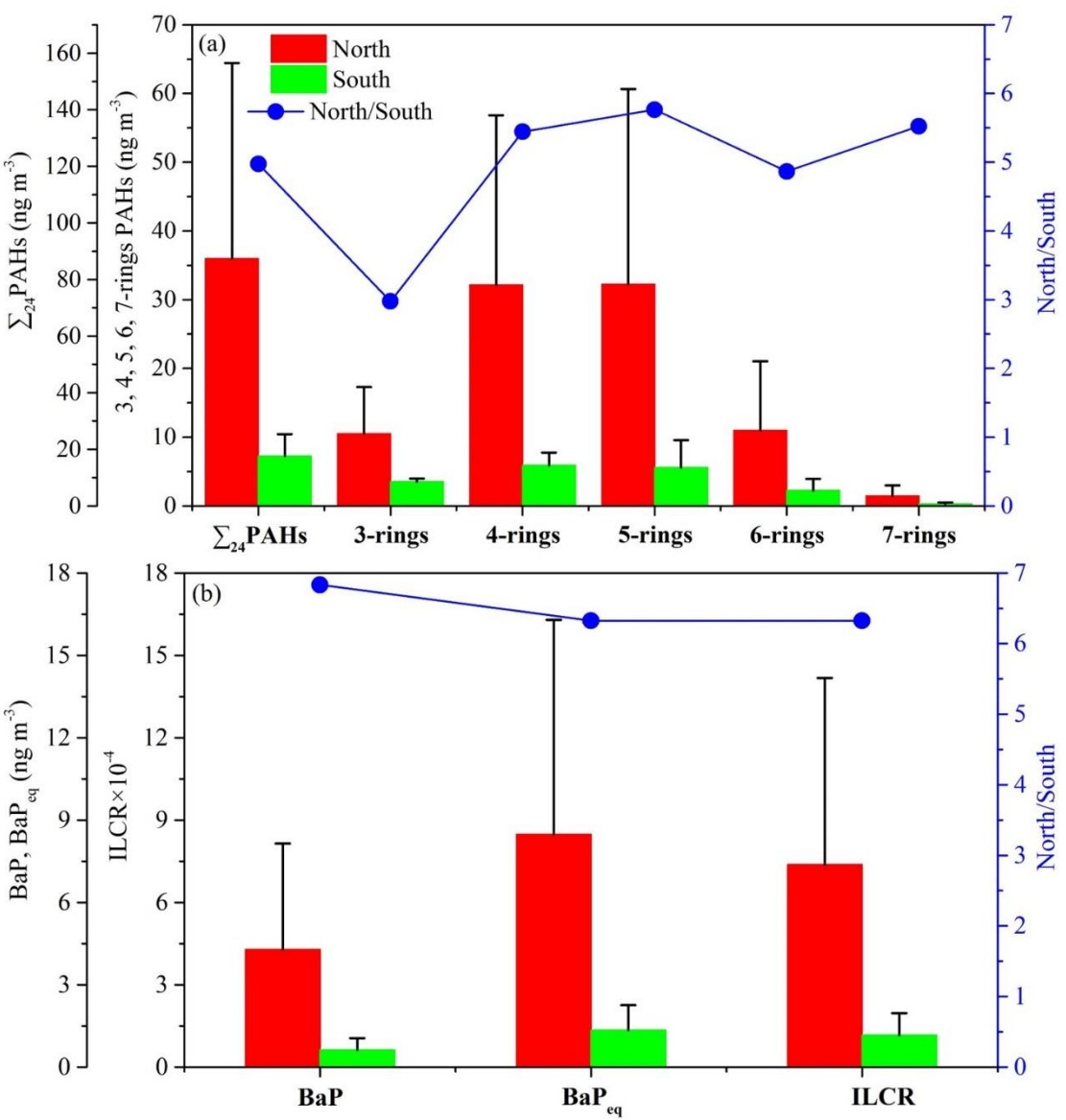


Figure 3 Comparison between the northern and the southern China in $\sum_{24}$PAHs, 3-7 rings PAHs
(a) and BaP, BaP$_{eq}$ and ILCR (b).

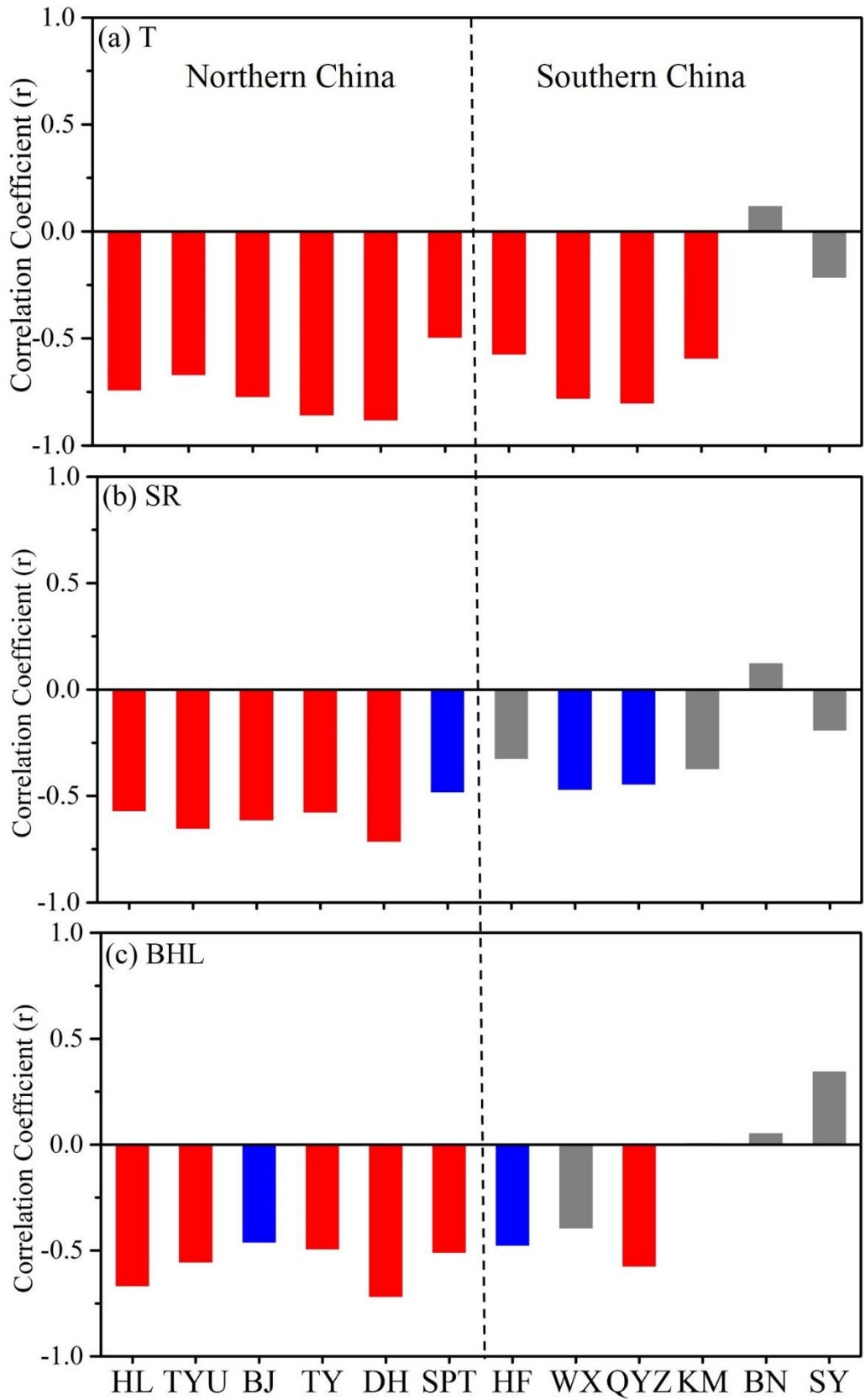


Figure 4 Correlation coefficient (r) of PAHs with T (a), SR (b) and BLH (c) at 12 sites. The

red, blue and gray bars indicate p<0.01, p<0.05 and p>0.05, respectively.


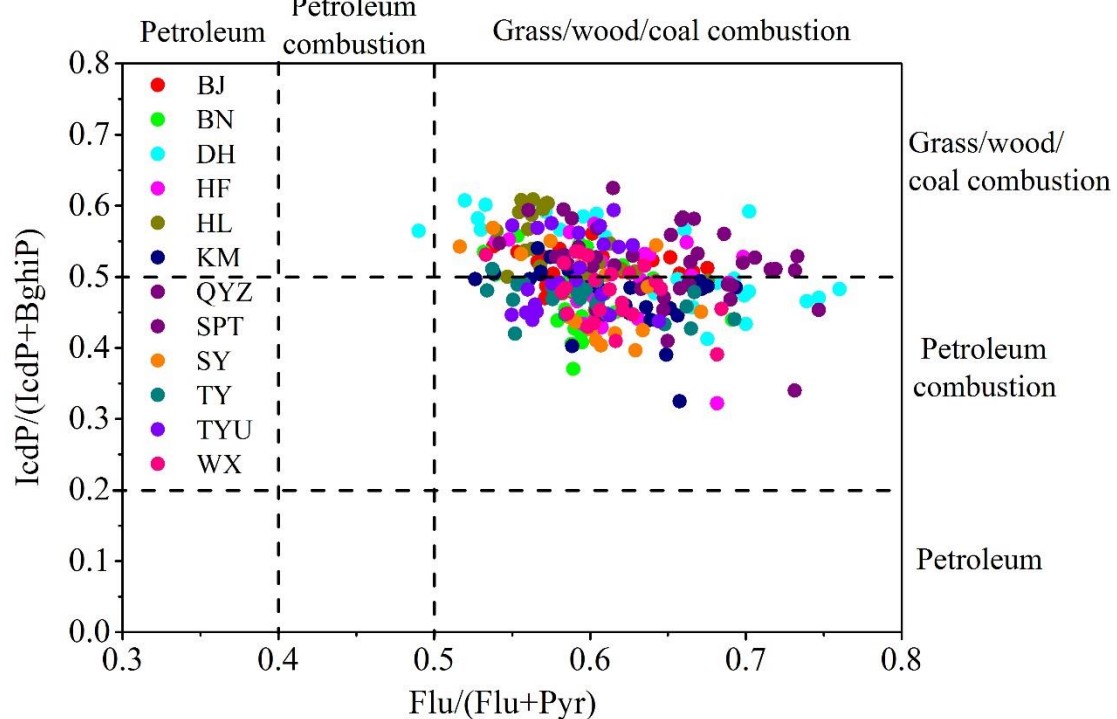


Figure 5 Diagnostic ratios of IcdP/(IcdP +BgiP) versus Flu/(Flu+Pyr) at 12 sites in China.

Ranges of ratios for sources are adopted from Yunker et al. (2002).

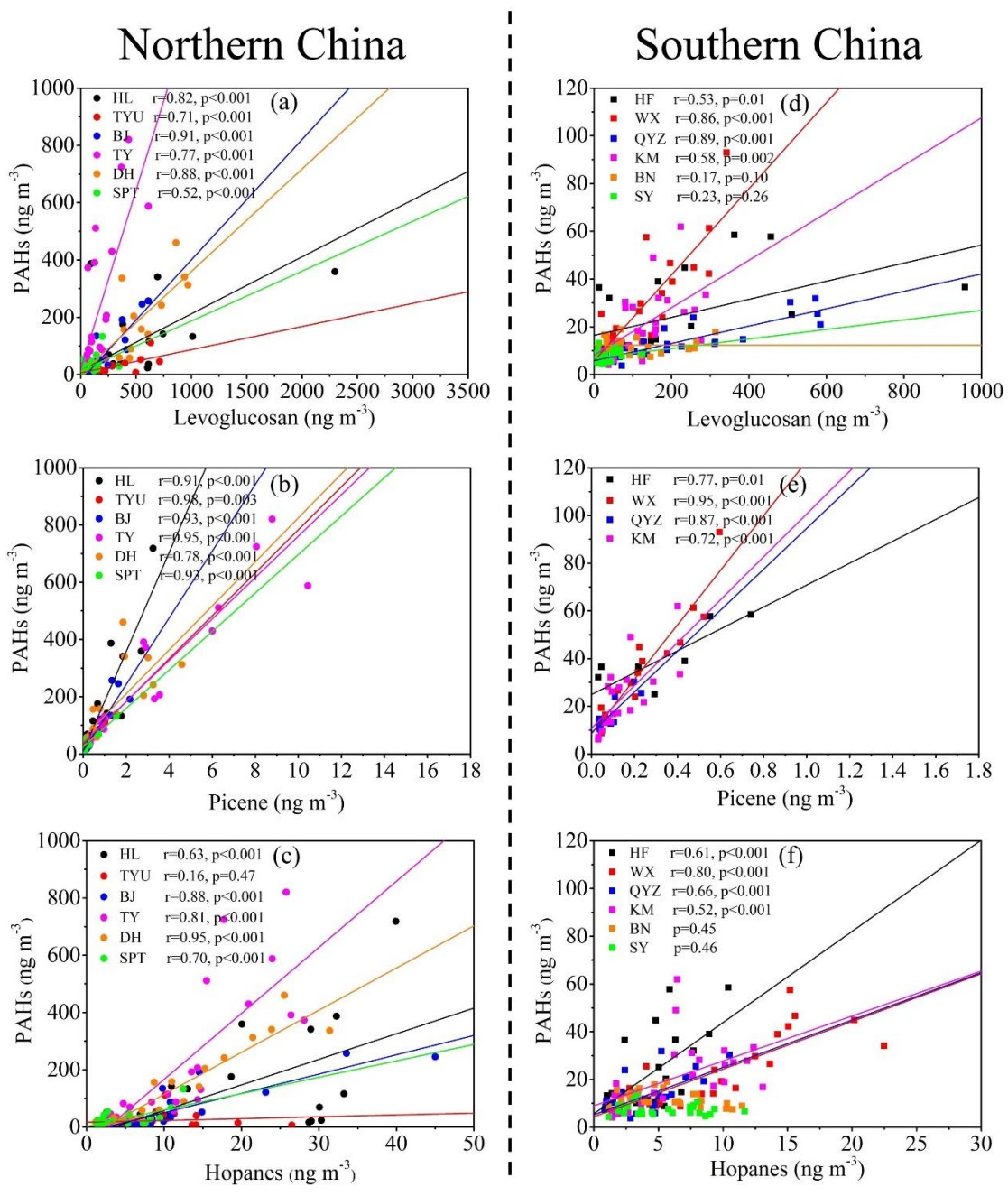


Figure 6 The correlation between PAHs and levoglucosan, picene and hopanes at sites in the

northern China (a-c) and the southern China (d-f).

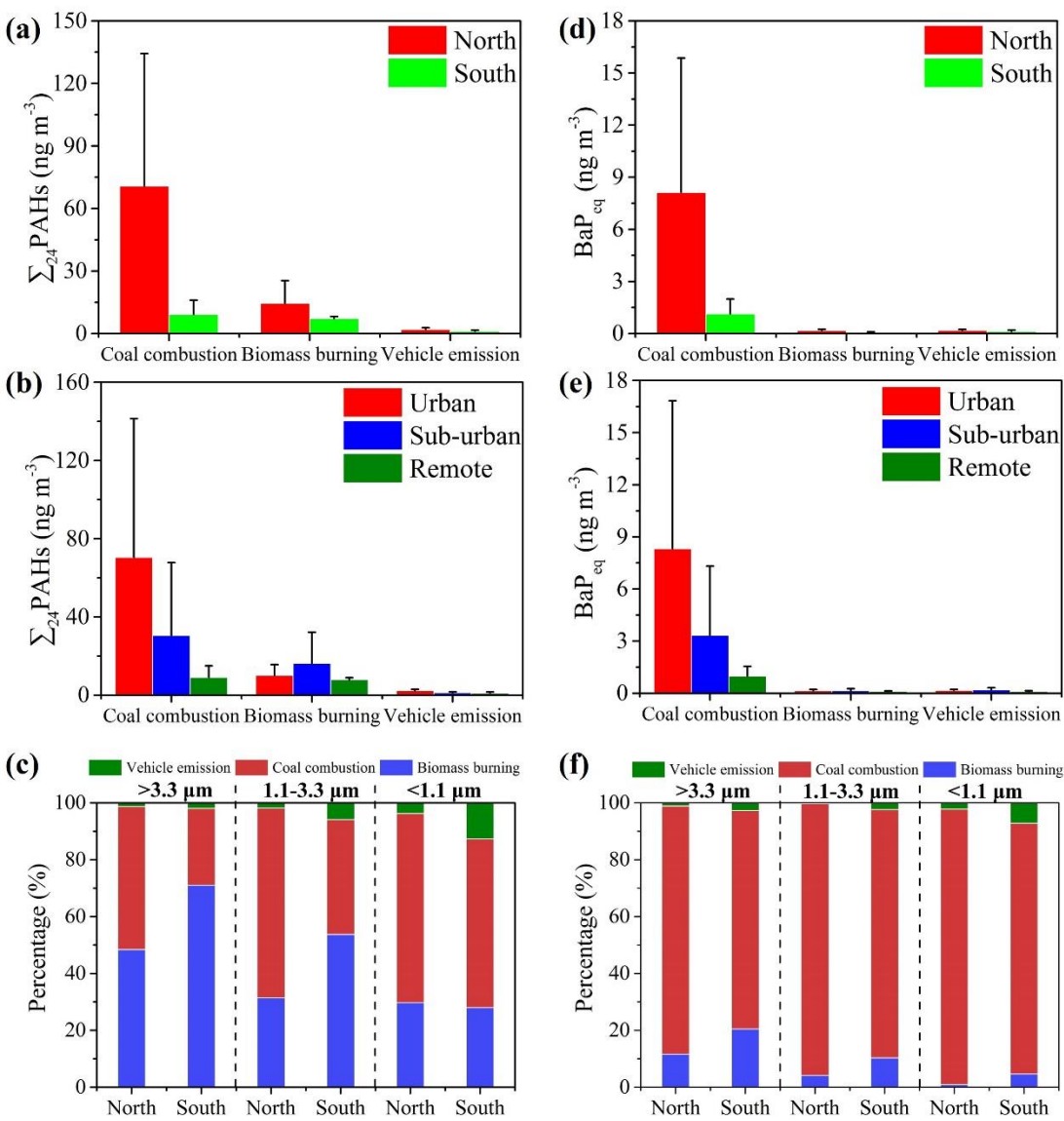


Figure 7 Source apportionment of $\sum_{24}$ PAHs and BaP$_{eq}$ in different regions (a, d), sampling sites
(b, e) and size particles (c, f).

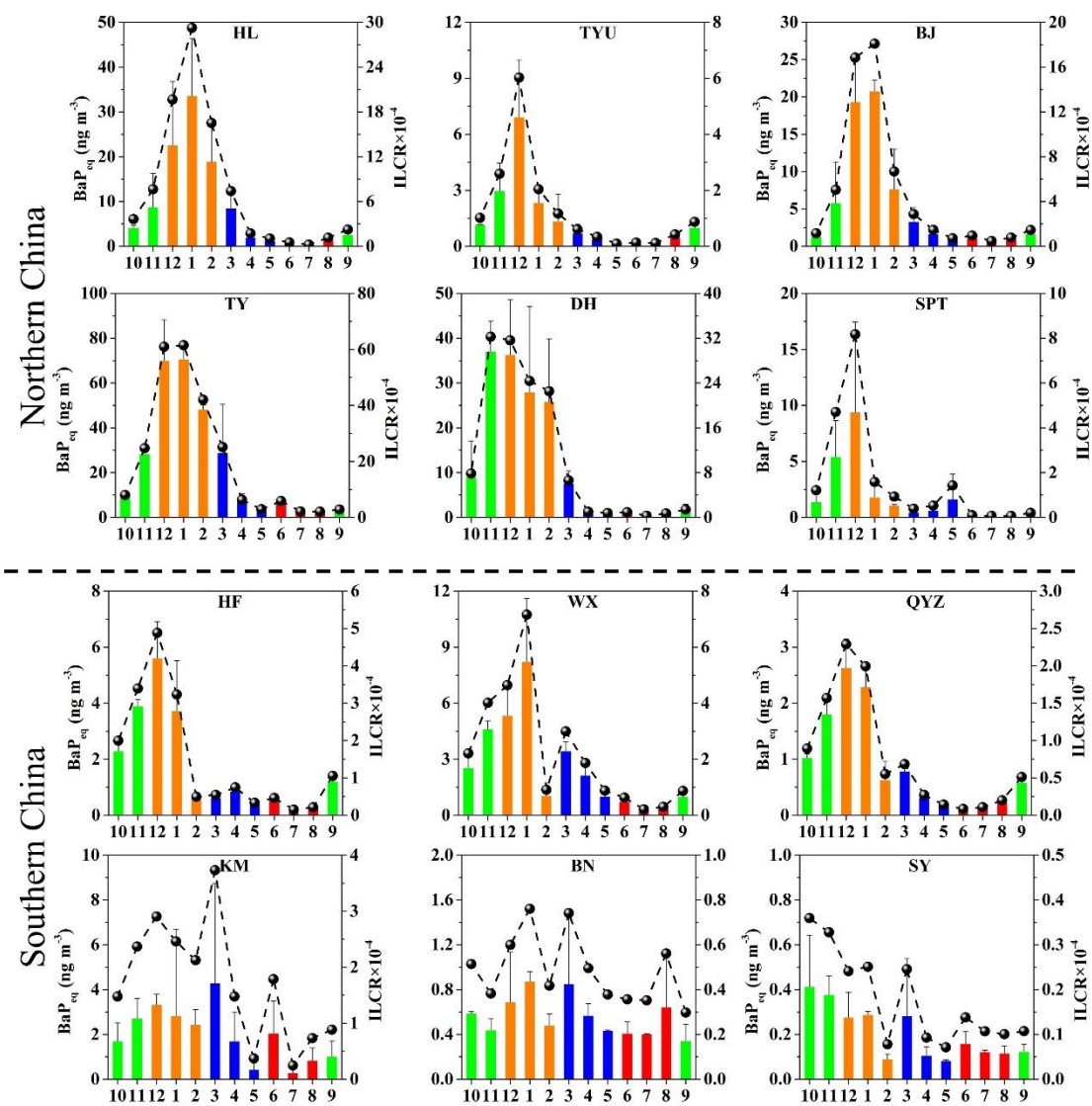


Figure 8 Monthly variations of BaP$_{eq}$ and ILCR at sites in the northern China and the southern

China. The green, yellow, blue and red bars represent BaP$_{eq}$ in fall (October − November, 2012

and September, 2013), winter (December 2012 – February 2013), spring (March – May, 2013),

and summer (June − August, 2013), respectively.

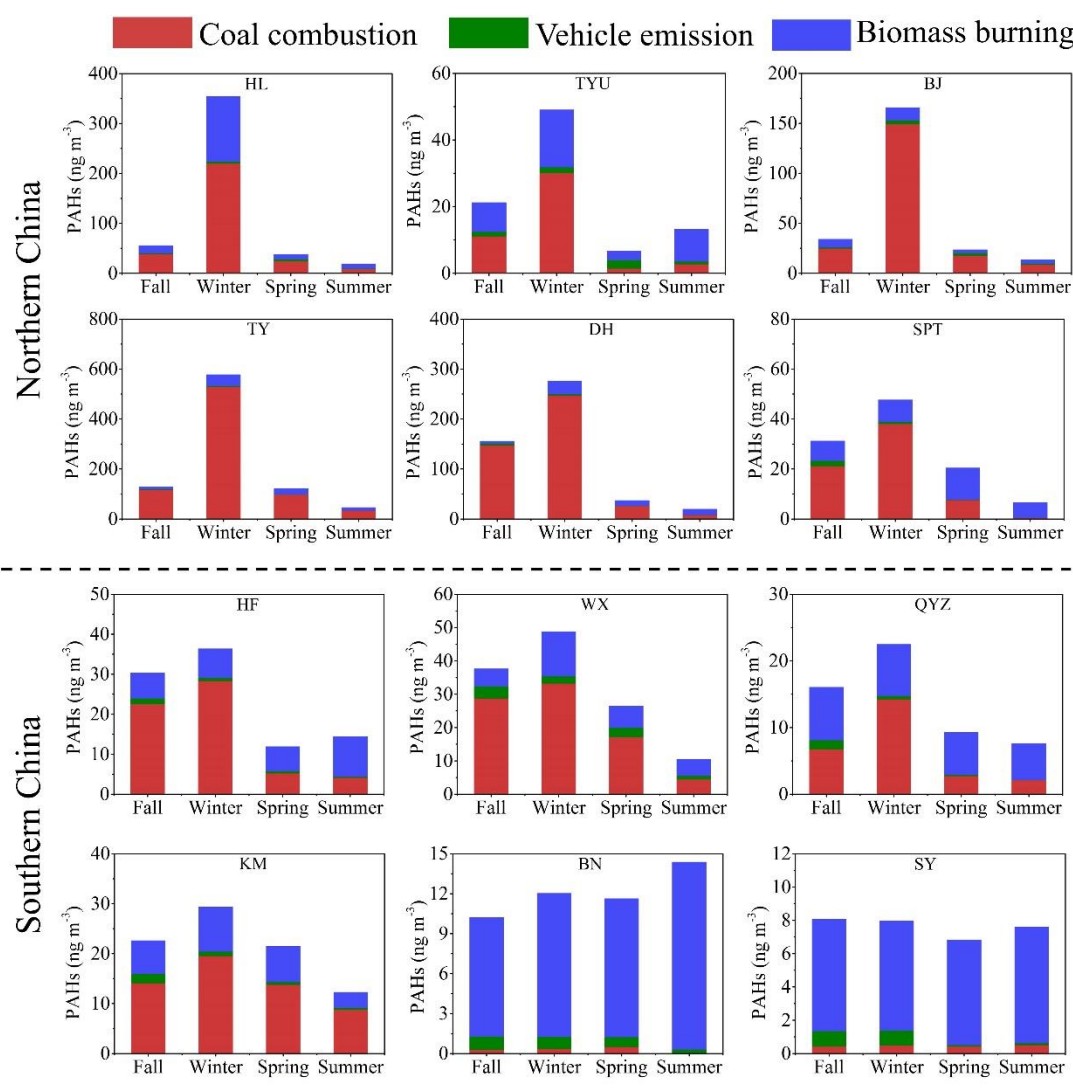

Figure 9 Seasonal variations of PAHs source contributions in China.

873