# Peer review of "Nationwide increase of polycyclic aromatic hydrocarbons in ultrafine particles during 2 winter over China revealed by size-segregated measurements"

_Atmospheric Chemistry and Physics, 2020_

## Short Comment (SC1) · 14 Jul 2020

Overall, this manuscript is short on bright spots and largely repeats known conclusions. The current level does not meet the publication standards of Atmospheric Chemistry and Physics.

I am not sure that using the same sampling and analysis methods to carry out TSP size-grading sampling in northern and southern China can be an advantage of this study. Because the results presented by the authors are comparable to those of other studies, this equivalence, to some extent, may indicate that the measurements are comparable despite the differences in sampling and analysis methods.

[Figure]

The correlation analysis between PAH concentration and meteorological parameters in this manuscript might be reconsidered. The meteorological parameters, T, SR, and BHL, were low in winter and high in summer, while the concentration of PAHs changed in the opposite way. This difference constitutes an inverse correlation between these meteorological parameters and the concentration of PAHs. Therefore, the authors' emphasis on the worsened PAH pollution in winter caused by adverse meteorological conditions is lack of argument. It is suggested to analyze the correlation between PAH concentration and meteorological parameters in northern and southern China in each season, and it is better to normalize the concentration at different sites. On the other hand, it is well known that the effect of meteorological conditions on pollutants is nonlinear. If feasible, it is desirable to use a nonlinear model to evaluate and even quantify the effect of meteorological conditions on the concentration of PAHs.

In addition, the source analysis of PAHs does not seem to be in-depth. It is expected to link the source contribution to the health risks of a specific PAH. This relation will improve the understanding of the impact of changes in emission sources on the composition and health risks of PAHs, which will be more conducive to the development of effective local control measures.

---

## Referee Comment (RC1) · Yingjun Chen (Referee) · 19 Jul 2020

The manuscript by Yu et al. reports one-year concurrent measurement of airborne PAHs at 12 sites across China. Size-segregated PAHs together with typical organic markers are measured to evaluate health risks of PAHs in different size particles and attribute emission sources of PAHs over different regions in China. The finding that toxic PAHs are concentrated in ultrafine particles is particularly interesting. The authors also find that PAH pollution is high in the northern China and nation-widely increases in wintertime, due to the unfavorable meteorological conditions and enhanced emissions of coal combustion and biomass burning. I think this is an important work nowadays in China as well in the global air pollution community. Overall this manuscript is well-organized and well-written and should be accepted after the authors address the minor issues below.

**Major comments:**

1. The PM samples were collected in 6 regions of China, including urban, sub-urban and remote sites. The authors are suggested to add more comparison of PAH concentrations and compositions among different types of sampling sites.

2. As I know, the national standard is not for $BaP_{eq}$ but BaP. The authors should directly compare measured BaP levels with the national standard.

**Specific comments:**

1. Line 52. Replace "associated" to "was associated".

2. Line 57. Replace "enriches" to "enrich".

3. Line 58. Replace "and" to "which".

4. Line 91. Delete "in".

5. Line 133. Replace "8h" to "8 h".

6. Line 146. Replace "3.3μm" to "3.3 μm".

7. Line 190. Replace "site" to "sites".

8. Line 214. The unit is misspelling. It should be "ng m$^{-3}$".

9. Line 259. Replace "high" to "higher".

10. Line 264. The abbreviation of boundary layer height is "BLH". Please replace "BHL" to "BLH" throughout the manuscript.

11. Line 281. Replace "within each northern region" to "within each region in the northern China".

12. Line 299. Replace "high" to "higher".

13. Line 306-308. The sentence "This is also confirmed by the significant correlations of $\sum_{24}$ PAHs with the biomass burning tracer, levoglucosan, the coal combustion tracer, picene, and the vehicle exhaust tracer, hopanes at most sites." should be re-phrased to "This is also confirmed by the significant correlations of $\sum_{24}$ PAHs with the typical tracers of biomass burning (levoglucosan), coal combustion (picene) and vehicle exhaust (hopanes)".

14. Line 314. Replace "biomass tracer" to "biomass burning tracer".

15. Line 338-340. Provide the full words for the abbreviation "SCE".

16. Figure 8. Please illustrate in the figure caption that the black dot-line represents the ILCR.

17. Table S4. Please add a line in the table to distinguish the sites in the northern China and the southern China.

18. Figure S11. Please add legend in the figure.

---

## Referee Comment (RC2) · Anonymous Referee #2 · 2 Aug 2020

This work conducted comprehensive field measurements of PAHs in fine particles at 12 sites in China for one year to investigate the chemical compositions, size-distributions, spatiotemporal variations, as well as the public health risk. In addition, diagnostic ratios and PMF model were applied to quantify the contributions from different sources to PAHs in northern China and sourthern China, highlighting the significant impacts from coal combustion and biomass burning, especially in winter in northern China. The manuscript is generally well written with clear logic, fluent language, abundant data, and deep analyses. There are some minor comments and suggestions below which are required to address before being accepted.

Specific comments:

1. Figure 1, keep the longitude and latitude of the map in same scale. If possible, try to use the coordinate of latitude and longitude instead of Cartesian coordinate when drawing the whole map of China.

2. Line 91, delete the extra word "in".

3. Line 134-135 and Figure 8, state the basis of season division. Why four months are included in summer but only two in autumn?

4. Line 149-153, point out the amount of the added internal standards and the specific extraction method.

5. Line 233-235 and 239-241, why most of the PAHs existed in ultrafine particles and the fractions in ultrafine particles varied with seasons? Is this related to the emission sources?

6. Section 3.3, is there any difference in the sources and contributions among urban, sub-urban and rural sites?

7. Line 336-340, the energy consumption data in 2008 from the Statistical Yearbook are not suitable for comparison. The data in 2013 can be used here.

8. Figure 8, it's better to remove the repeated ordinate title of the middle graphs.

---

## Author Comment (AC1) · 29 Sep 2020

**Comment on manuscript on acp-2020-576**

Overall, this manuscript is short on bright spots and largely repeats known conclusions. The current level does not meet the publication standards of Atmospheric Chemistry and Physics.

I am not sure that using the same sampling and analysis methods to carry out TSP size-grading sampling in northern and southern China can be an advantage of this study. Because the results presented by the authors are comparable to those of other studies, this equivalence, to some extent, may indicate that the measurements are comparable despite the differences in sampling and analysis methods.

Reply: Thank you for your comment. Public concerns on polycyclic aromatic hydrocarbons (PAHs) are mainly due to their carcinogenic potential. As the largest developing country in the world, China is the largest PAHs emitter and has high cancer risks caused by PAHs exposure. PAHs in different size particles have different health impacts. Thus, it is essential to understand size distribution of PAHs levels and sources and discover their difference in health risks among typical regions of China (e.g. north vs. south, urban vs. remote). These results are helpful to provide a basis for PAHs pollution control and health effects reduction in different regions of China. Unfortunately, most previous studies on atmospheric PAHs are undertaken at several sites within a local or regional scale in China. Due to the inconsistency in sampling methods, frequency and duration in these local and regional campaigns, it is difficult to draw a national picture of PAHs pollution in the air of China. To the best of our knowledge, our national observation is one of the first studies to acquire comprehensive information concerning spatiotemporal characteristics, source apportionment and health risks of size-segregated PAHs over a large national scale.

Based on our observation, we find that PAHs and $BaP_{eq}$ are dominated in $PM_{1.1}$ at all sites,

indicating that high carcinogenicity of PAHs is accompanied with ultrafine particles. Nationwide increases in both PAH levels and inhalation cancer risks occur in winter, probably due to the unfavorable meteorological conditions and enhanced emissions of coal combustion and biomass burning. Moreover, in the revised manuscript, we add more discussion focusing on PAHs and $BaP_{eq}$ sources in different size particles and among urban, sub-urban and remote sites. We find that coal combustion is the major source of $BaP_{eq}$ in all size particles at most monitoring sites. We believe that these findings provide insights into PAHs pollution and its potential effect on public health in China. Thus, this information is helpful to provide a basis for PAHs pollution control and health effects reduction in different regions of China.

The correlation analysis between PAH concentration and meteorological parameters in this manuscript might be reconsidered. The meteorological parameters, T, SR, and BHL, were low in winter and high in summer, while the concentration of PAHs changed in the opposite way. This difference constitutes an inverse correlation between these meteorological parameters and the concentration of PAHs. Therefore, the authors' emphasis on the worsened PAH pollution in winter caused by adverse meteorological conditions is lack of argument. It is suggested to analyze the correlation between PAH concentration and meteorological parameters in northern and southern China in each season, and it is better to normalize the concentration at different sites. On the other hand, it is well known that the effect of meteorological conditions on pollutants is nonlinear. If feasible, it is desirable to use a nonlinear model to evaluate and even quantify the effect of meteorological conditions on the concentration of PAHs.

Reply: Thank you for your suggestion. Theoretically, adverse meteorological conditions (low

temperature, solar radiation and boundary layer height, etc.) indeed lead to the increase of particulate PAHs. PAHs are semi-volatile compounds (SVOCs) and can partition between the gas and particle phases. The gas-particle (G/P) partitioning behavior of atmospheric PAHs can be described as equations (1) and (2) (Pankow, 1994).

$$K_{p,OM} = \frac{RT}{10^6 \overline{MW_{OM}} \zeta_{OM} P_L^o} \qquad (1)$$

$$P_L^o = P_L^{o,*} \exp\left[\frac{\Delta H_{vap}^*}{R} \left(\frac{1}{298.15} - \frac{1}{T}\right)\right] \qquad (2)$$

where $K_{p,OM}$ represents the absorptive G/P partitioning coefficient of individual PAH, R (m$^3$ Pa/(K/mol)) is the ideal gas constant, T (K) is the ambient temperature. $\overline{MW_{OM}}$ (g/mol) is the mean molecular weight of organic matter (OM) and is assumed to be 200 g/mol (Xie et al., 2014), $\zeta_{OM}$ is the scale activity coefficient of each compound in the absorbing phase and is usually assumed to be unity. $P_L^{o,*}$ is the vapor pressure of each PAH at 298.15K and $\Delta H_{vap}^*$ is vaporization enthalpy of the liquid at 298.15K. Thus, for a specific PAH in a single OM phase at a fixed relative humidity, the G/P partitioning should be driven by ambient temperature only. As Figure 1 showed, the decrease of ambient temperature can cause the increase of $K_{p,OM}$. This means that the decrease of ambient temperature would result in the increase of individual PAH in the particulate phase assuming a constant total concentration in the air.

[Figure]

Figure 1 The $K_{p, OM}$ ($m^3$ $ug^{-1}$) under different temperature.

In the atmosphere, PAHs removal by OH can be described as:

$$\frac{dC_{PAH}}{dt} = -k * [OH] * C_{PAH} \qquad (3)$$

where k is the rate constant for the reaction of a PAH with OH radical, $C_{PAH}$ is the concentration of individual PAH in the air. Solar radiation (SR) directly affects photochemistry in the air. As Figure 2 showed, solar radiation values during our campaign positively correlated with the concentrations of hydroxyl radical [OH] which were estimated based on the empirical equation (4) (Ehhalt and Rohrer, 2000). Thus, the decrease of SR can indeed lower [OH] and accumulate PAHs in the air, resulting in the increase of PAHs concentrations.

$$[OH] = a(JO^1D)^\alpha (JNO_2{}^\beta) \frac{bNO_2 + 1}{cNO_2{}^2 + dNO_2 + 1} \qquad (4)$$

[Figure]

Figure 2 Correlation between OH concentration and solar radiation.

For the influence of boundary layer, low height of boundary layer can inhibit the vertical diffusion of PAHs, which leads to PAHs accumulation and increased concentrations.

We agree with the reviewer that the effect of meteorological conditions on pollutants is nonlinear. It is better to use a nonlinear model to evaluate the effect, which is out of the scope of the current study. At least above discussion illustrates theoretical inverse relationships between these meteorological parameters (temperature, solar radiation and boundary layer height) and the concentration of particulate-bound PAHs.

As suggested by the reviewer, we try to analyze the correlations between PAH concentrations and meteorological parameters in each season. Unfortunately there is only six samples in each season at a site. Instead, we divide the one-year data into warm and cold seasons based on the ambient temperature. As Figure 3 showed, at most sites in the northern and southern China, PAHs negatively correlated with temperature (T), boundary layer height (BLH) and solar radiation (SR) in both cold (T < 10 °C) and warm (T > 10 °C) seasons. Thus, coupled with above theoretical discussion, we believe our correlation analysis does reflect the effect of meteorological parameter on PAH concentrations.

In the revised manuscript, we add more discussion about the effect of meteorological parameter on PAH concentrations in Line 297-305 and Line 318-323. And Figure 3 was added in supporting information file and the revised manuscript as Figure S10. The detail theoretical discussion information had been added to the supporting information as Text S1 (The line numbers here refers to the 'tracking changes' file)

[Figure]

Figure 3 Correlation coefficient (r) of PAHs with T (a), SR (b) and BHL (c) at 12 sites in cold and warm season.

*: $p < 0.05$

**: the ambient temperature in KM, BN and SY are all exceed ten degree, there are no cold season in these three sampling sites.**

In addition, the source analysis of PAHs does not seem to be in-depth. It is expected to link the source contribution to the health risks of a specific PAH. This relation will improve the understanding of the impact of changes in emission sources on the composition and health risks of PAHs, which will be more conducive to the development of effective local control measures.

Reply: Thank you for your suggestion. BaP carcinogenic equivalent concentration ($BaP_{eq}$) is widely used to evaluate the health risks of PAHs. In the revised manuscript, we add more discussion focusing on source apportionment of $BaP_{eq}$ as well as $\sum_{24}PAHs$ in different size particles and urban, sub-urban and remote sites. To the best of our knowledge, this is one of the first studies to acquire comprehensive information concerning observation-based source apportionment of size-segregated PAHs and $BaP_{eq}$ over a large national scale.

Figure 4 show source apportionment of $\sum_{24}PAHs$ in different regions (a), sampling sites (b) and size particles (c). In the northern China, coal combustion was the major source of atmospheric PAHs (73.6 ng m$^{-3}$, 84.2% of $\sum_{24}PAHs$), followed by biomass burning (11.8 ng m$^{-3}$ and 13.5%) and vehicle exhaust (2.0 ng m$^{-3}$ and 2.3%). In the southern China, coal combustion (9.6 ng m$^{-3}$ and 54.8%) and biomass burning (6.8 ng m$^{-3}$ and 39.0%) were the major contributors, followed by vehicle exhaust (1.1 ng m$^{-3}$ and 6.2%) (Figure 4a). At urban and sub-urban sites, coal combustion was the largest source of $\sum_{24}PAHs$ (70.4 ng m$^{-3}$, 85.1% and 30.5 ng m$^{-3}$, 63.5%), followed by biomass burning (10.1 ng m$^{-3}$, 12.2% and 16.3 ng m$^{-3}$, 33.9%) and vehicle emission (2.2 ng m$^{-3}$, 2.6% and 1.2 ng m$^{-3}$, 2.5%), while at remote sites the contributions of coal combustion (9.1 ng m$^{-3}$, 50.6% ) and biomass burning (7.8 ng m$^{-3}$, 43.7%) were comparable and vehicle emission (1.0 ng m$^{-3}$, 5.7%) had minor contributions. The major sources of $\sum_{24}PAHs$ varied among different size particles in the northern and southern China (Figure 4c).

For PM$_{>3.3}$-bound PAHs, the contributions of coal combustion (50.3%) and biomass burning (48.4%) were comparable in the northern China, while biomass burning (71.0%) was the largest source in the southern China. For PM$_{1.1-3.3}$-bound PAHs, coal combustion (66.7%) was the dominated source in the northern China, whereas the percentage of biomass burning (53.7%) was larger than that of coal combustion (40.4%) in the southern China. For PM$_{1.1}$-bound PAHs, coal combustion was the dominated source in the northern (66.6%) and southern (59.3%) China.

Figure 4 shows source apportionment of BaP$_{eq}$ in different regions (d), sampling sites (e) and size particles (f). Unlike $\sum_{24}$PAHs, coal combustion was the predominant source of BaP$_{eq}$ in the northern (8.1 ng m$^{-3}$ and 95.7%) and the southern (1.1 ng m$^{-3}$ and 84.7%) China. The contributions of coal contribution at urban sites (8.3 ng m$^{-3}$ and 96.4%) were larger than those at sub-urban (3.3 ng m$^{-3}$ and 90.8%) and remote (1.0 ng m$^{-3}$ and 82.5%) sites. Coal combustion was the dominate source in different size particles. And its contributions to PM$_{>3.3}$, PM$_{1.1-3.3}$ and PM$_{1.1}$-bound PAHs in the northern China (87.3%, 95.6% and 96.9%) were all larger than those in the southern China (76.8%, 87.3% and 88.2%).

All these discussion has been added to the revised manuscript in Line 381-386 and Line 409-430. And Figure 4 was added in the revised manuscript as Figure 7. We believe these findings provide insights into the linkage between the source contributions to the health risks of atmospheric PAHs and improve the understanding of the impact of changes in emission sources on the compositions and health risks of PAHs.

[Figure]

Figure 4 Source apportionment of $\sum_{24}$ PAHs and BaP$_{eq}$ in different regions (a, c), sampling sites (b, d) and size particles (c, f).

References

Ehhalt, D.H., Rohrer, F., 2000. Dependence of the OH concentration on solar UV. J. Geophys. Res.-Atmos. 105, 3565-3571.

Pankow, J.F., 1994. An absorption model of gas/particle partitioning of organic compounds in the atmosphere. Atmos. Environ. 28, 185-188.

Xie, M., Hannigan, M.P., Barsanti, K.C., 2014. Gas/particle partitioning of n-alkanes, PAHs

and oxygenated PAHs in urban Denver. Atmos. Environ. 95, 355-362.

---

## Author Comment (AC2) · 29 Sep 2020

**Referee comment to acp-2020-576-RC1**

The manuscript by Yu et al. reports one-year concurrent measurement of airborne PAHs at 12 sites across China. Size-segregated PAHs together with typical organic markers are measured to evaluate health risks of PAHs in different size particles and attribute emission sources of PAHs over different regions in China. The finding that toxic PAHs are concentrated in ultrafine particles is particularly interesting. The authors also find that PAH pollution is high in the northern China and nation-widely increases in wintertime, due to the unfavorable meteorological conditions and enhanced emissions of coal combustion and biomass burning. I think this is an important work nowadays in China as well in the global air pollution community. Overall this manuscript is well-organized and well-written and should be accepted after the authors address the minor issues below.

Major comments:

1. The PM samples were collected in 6 regions of China, including urban, sub-urban and remote sites. The authors are suggested to add more comparison of PAH concentrations and compositions among different types of sampling sites.

Reply: Thank you for your suggestion. In the revised manuscript, we add more discussion on $\sum_{24}$PAHs concentrations, compositions, sources and $BaP_{eq}$ concentration and sources among different types of sampling sites. And Figure 1-4 were added in supporting information file and the revised manuscript as Figure S2, Figure S4, Figure S5 and Figure S6. Figure 5 and Figure 6 was added in the revised manuscript as Figure 7b and Figure 7e. (The line numbers here refers to the 'tracking changes' file)

"The concentrations of $\sum_{24}$PAHs at urban sites (82.7 ng m$^{-3}$) were significant higher (p<0.05)

than those at sub-urban (48.0 ng m$^{-3}$) and remote sites (18.0 ng m$^{-3}$) (Figure 1) (Line 228-229).

And BeP$_{eq}$ (Figure 2) and ILCR (Figure 3) were both the highest at urban sites. All these indicated that people in urban regions of China were faced with higher exposure risk of PAHs pollution as compared to those in rural and remote areas. Figure 4 exhibits that 4- and 5-rings PAHs are the majority in $\sum_{24}$PAHs at urban, sub-urban and remote sites, which totally accounted 72.2, 63.8 and 66.6% of the total amounts in TSP, respectively. The percentage of 5-rings PAHs dominates at urban sites, and 4-rings PAHs makes the largest proportion at sub-urban and remote sites (Line 244-250). PMF result showed that at urban and sub-urban sites coal combustion was the largest source of $\sum_{24}$PAHs (70.4 ng m$^{-3}$, 85.1% and 30.5 ng m$^{-3}$, 63.5%), followed by biomass burning (10.1 ng m$^{-3}$, 12.2% and 16.3 ng m$^{-3}$, 33.9%) and vehicle emission (2.2 ng m$^{-3}$, 2.6% and 1.2 ng m$^{-3}$, 2.5%), while at remote sites the contributions of coal combustion (9.1 ng m$^{-3}$, 50.6% ) and biomass burning (7.8 ng m$^{-3}$, 43.7%) were comparable and vehicle emission (1.0 ng m$^{-3}$, 5.7%) had minor contributions (Figure 5) (Line 410-415 ). Coal combustion was the predominated source of BaP$_{eq}$, and its contribution at urban sites (8.3 ng m$^{-3}$ and 96.4%) were larger than those at sub-urban (3.3 ng m$^{-3}$ and 90.8%) and remote (1.0 ng m$^{-3}$ and 82.5%) sites. (Figure 6)" (Line 426-428)

[Figure]

Figure 1 Concentrations of $\sum_{24}$PAHs at urban, sub-urban and remote sites.

[Figure]

Figure 2 Concentrations of BaP$_{eq}$ at urban, sub-urban and remote sites.

[Figure]

Figure 3 ILCR at urban, sub-urban and remote sites.

[Figure]

Figure 4 PAHs composition at urban, sub-urban and remote sites.

[Figure]

Figure 5 Difference of $\sum_{24}$PAHs sources at urban, sub-urban and remote sites.

[Figure]

Figure 6 Difference of BaP$_{eq}$ sources at urban, sub-urban and remote sites.

2. As I know, the national standard is not for BaP$_{eq}$ but BaP. The authors should directly compare measured BaP levels with the national standard.

Reply: Yes, the national standard (1.0 ng m$^{-3}$) is for BaP. In the revised manuscript, we directly compare measured BaP levels with the national standard.

"Annual averages of BaP in TSP among the 12 sites were in the range of 0.09 to 11.0 ng m$^{-3}$ with a mean of 2.58 ng m$^{-3}$. The highest level of atmospheric BaP occurred at TY and the lowest existed at SY. The BaP values at five sites (WX, BJ, HL, DH and TY) exceeded the national standard of annual atmospheric BaP (1.0 ng m$^{-1}$) by factors of 1.2 to 11.0. For BaP$_{eq}$, annual averages ranged from 0.21 to 22.2 ng m$^{-3}$ with the predominant contribution from 5-rings PAHs (Figure 1b)." (Line 230-235)

Specific comments:

1. Line 52. Replace "associated" to "was associated".

Reply: Revised as suggested. (Line 57)

2. Line 57. Replace "enriches" to "enrich".

Reply: Revised as suggested. (Line 62)

3. Line 58. Replace "and" to "which".

Reply: Revised as suggested. (Line 63)

4. Line 91. Delete "in".

Reply: Revised as suggested. (Line 96)

5. Line 133. Replace "8h" to "8 h".

Reply: Revised as suggested. (Line 144)

6. Line 146. Replace "3.3μm" to "3.3 μm".

Reply: Revised as suggested. (Line 161)

7. Line 190. Replace "site" to "sites".

Reply: Revised as suggested. (Line 207)

8. Line 214.The unit is misspelling. It should be "ng m $^{-3}$".

Reply: Revised as suggested.

9. Line 259. Replace "high" to "higher".

Reply: Revised as suggested. (Line 294)

10. Line 264. The abbreviation of boundary layer height is "BLH". Please replace "BHL" to

"BLH" throughout the manuscript.

Reply: Revised as suggested. (Line 307, Line 309, Line 311, Line 316, Line 446, Line 788)

11. Line 281. Replace "within each northern region" to "within each region in the northern China".

Reply: Revised as suggested. (Line 330-331)

12. Line 299. Replace "high" to "higher".

Reply: Revised as suggested. (Line 349)

13. Line 306-308. The sentence "This is also confirmed by the significant correlations of $\sum_{24}$PAHs with the biomass burning tracer, levoglucosan, the coal combustion tracer, picene, and the vehicle exhaust tracer, hopanes at most sites." should be re-phrased to "This is also confirmed by the significant correlations of $\sum_{24}$ PAHs with the typical tracers of biomass burning (levoglucosan), coal combustion (picene) and vehicle exhaust (hopanes)".

Reply: Revised as suggested. (Line 356-357)

14. Line 314. Replace "biomass tracer" to "biomass burning tracer".

Reply: Revised as suggested. (Line 367)

15. Line 338-340. Provide the full words for the abbreviation "SCE".

Reply: Revised as suggested. (Line 391-392)

16. Figure 8. Please illustrate in the figure caption that the black dot-line represents the ILCR.

Reply: Revised as suggested.

17. Table S4. Please add a line in the table to distinguish the sites in the northern China and the southern China.

Reply: We revised Table S4 to distinguish the sites in the northern China and the southern China.

18. Figure S11. Please add legend in the figure.

Reply: Revised as suggested.

---

## Author Comment (AC3) · 29 Sep 2020

**Referee comment to acp-2020-576-RC2**

This work conducted comprehensive field measurements of PAHs in fine particles at 12 sites in China for one year to investigate the chemical compositions, size-distributions, spatiotemporal variations, as well as the public health risk. In addition, diagnostic ratios and PMF model were applied to quantify the contributions from different sources to PAHs in northern China and southern China, highlighting the significant impacts from coal combustion and biomass burning, especially in winter in northern China. The manuscript is generally well written with clear logic, fluent language, abundant data, and deep analyses. There are some minor comments and suggestions below which are required to address before being accepted.

Specific comments:

1. Figure 1, keep the longitude and latitude of the map in same scale. If possible, try to use the coordinate of latitude and longitude instead of Cartesian coordinate when drawing the whole map of China.

Reply: Thank you for your suggestion. We add the coordinate of latitude and longitude in Figure 1.

[Figure]

Figure 1 Annual averages of $\sum_{24}$ PAHs (a) and BaP$_{eq}$ (b) at 12 sites in China.

2. Line 91, delete the extra word "in".

Reply: Revised as suggested. (Line 96) (The line numbers here refers to the 'tracking changes'

file).

3. Line 134-135 and Figure 8, state the basis of season division. Why four months are included in summer but only two in autumn?

Reply: Season division is based on consistent annual changes in the weather. According to the meteorological definition, each season lasts three months that spring runs from March to May, summer runs from June to August, fall (autumn) runs from September to November, and winter runs from December to February. In the revised manuscript, we state the basis of season division in the caption and revise the figure. Figure 2 here was Figure 8 in the revised manuscript.

[Figure]

Figure 2 Monthly variations of BaP$_{eq}$ and ILCR at sites in the northern China and the southern

China. The green, yellow, blue and red bars represent BaP$_{eq}$ in fall (October-November, 2012 and September, 2013), winter (December 2012-February 2013), spring (March-May, 2013), and summer (June-August, 2013), respectively. The black dot represented the ILCR.

4. Line 149-153, point out the amount of the added internal standards and the specific extraction method.

Reply: Thank you for your suggestion. In this study, we added 400 μL of internal standards into each sample. The extraction method is ultrasonic solvent extraction.

"Before ultrasonic solvent extraction, 400 μL of isotope-labeled mixture compounds (tetracosane-d50, napthalene-d8, acenaphthene-d10, phenanthrene-d10, chrysene-d12, perylene-d12 and levoglucosan-$^{13}$C6) were spiked into the samples as internal standards." (Line 161-164)

5. Line 233-235 and 239-241, why most of the PAHs existed in ultrafine particles and the fractions in ultrafine particles varied with seasons? Is this related to the emission sources?

Reply: Yes, it should be related to the emission sources of PAHs. Atmospheric PAHs are mainly derived from combustion sources. As Shen et al. (2013) reported (see Figure 3 below), PAHs emitted form biomass burning and coal combustion enriched in ultrafine particles (<1.1 μm). Moreover, coal combustion witnessed more enrichment of PAHs in ultrafine particles than biomass burning.

[Figure]

Figure 3 Size distribution of particle-phase PAHs in emissions for different fuels (Shen et al., 2013)

Our PMF results showed an apparently seasonal trend of PAHs sources. For instant, at the DH site, the contributions of coal combustion kept decreasing from winter to summer, while biomass burning kept increasing (Figure 4a). Such a change in PAH sources indeed resulted in the seasonal variations of PAH fractions in ultrafine particles that the mass fractions of $\Sigma_{24}$PAHs in $PM_{1.1}$ were the highest during fall to winter and the lowest during summer (Figure 4b).

[Figure]

Figure 4 Seasonal variations of PAHs source contributions at DH (a); size distribution of

$\sum_{24}$PAHs in different season at DH.

Shen, G.F., Tao, S., Chen, Y.C., Zhang, Y.Y., Wei, S.Y., Xue, M., Wang, B., Wang, R., Lu, Y., Li, W., Shen, H.Z., Huang, Y., Chen, H., 2013. Emission characteristics for polycyclic aromatic hydrocarbons from solid fuels burned in domestic stoves in rural China. Environ. Sci. Technol. 47, 14485-14494.

6. Section 3.3, is there any difference in the sources and contributions among urban, sub-urban and rural sites?

Reply: Figure 5 and Figure 6 show the difference of $\sum_{24}$PAHs and BaP$_{eq}$ sources at urban, sub-urban and rural sites, respectively. At urban and sub-urban sites, coal combustion was the largest source of $\sum_{24}$PAHs, followed by biomass burning and vehicle emission, while at remote sites the contributions of coal combustion and biomass burning were comparable and vehicle emission had minor contributions. Coal combustion was the predominated source of BaP$_{eq}$, and its contribution at urban sites were larger than those at sub-urban and remote sites.

In the revised manuscript, we add more discussion on the difference in the $\sum_{24}$PAH and BaP$_{eq}$ sources among urban, sub-urban and rural sites in Line 410-415 and Line 426-428. Figure 5 and Figure 6 was added in the revised manuscript as Figure 7b and Figure 7e.

"At urban and sub-urban sites coal combustion was the largest source of $\sum_{24}$PAHs (70.4 ng m$^{-3}$, 85.1% and 30.5 ng m$^{-3}$, 63.5%), followed by biomass burning (10.1 ng m$^{-3}$, 12.2% and 16.3 ng m$^{-3}$, 33.9%) and vehicle emission (2.2 ng m$^{-3}$, 2.6% and 1.2 ng m-3, 2.5%), while at remote sites the contributions of coal combustion (9.1 ng m$^{-3}$, 50.6% ) and biomass burning (7.8 ng m$^{-3}$, 43.7%) were comparable and vehicle emission (1.0 ng m$^{-3}$, 5.7%) had minor contributions (Figure 5) (Line 410-415). Coal combustion was the predominated source of BaP$_{eq}$, and its

contribution at urban sites (8.3 ng m$^{-3}$ and 96.4%) were larger than those at sub-urban (3.3 ng m$^{-3}$ and 90.8%) and remote (1.0 ng m$^{-3}$ and 82.5%) sites. (Figure 6) (Line 426-428)

[Figure]

Figure 5 Difference of $\sum_{24}$PAHs sources at urban, sub-urban and remote sites.

[Figure]

Figure 6 Difference of BaP$_{eq}$ sources at urban, sub-urban and remote sites.

7. Line 336-340, the energy consumption data in 2008 from the Statistical Yearbook are not suitable for comparison. The data in 2013 can be used here.

Reply: Thank you for your suggestion. The data in 2013 from the Statistical Yearbook was used in the revised manuscript. (see below)

"As China statistics yearbook recorded (http://www.stats.gov.cn/english/Statisticaldata/AnnualData/), coal was the dominant fuel in China, accounting for 70.6% (24.1×10$^8$ tons of Standard Coal Equivalent, SCE) of total primary energy consumption (34.1×10$^8$ tons of SCE) in 2012, followed by crude oil 19.9% (6.7×10$^8$ tons of SCE) and other types of energy 9.5%, including biofuel, natural gas, hydro power, nuclear power and other power (3.2×10$^8$ tons of SCE)." (Line 390-395)

8. Figure 8, it's better to remove the repeated ordinate title of the middle graphs.

Reply: Revised as suggested.

---

## Author Response (AR2)

Dear Editor,

Thank you very much for handling our manuscript (MS No.: acp-2020-576) submitted to Atmospheric Chemistry and Physics.

We have adequately addressed all the comments raised, and revised our manuscript accordingly again. For more information, please refer to our revised manuscript and replies to referees.

I would like to thank you and referees for all your efforts, which have largely help us improve our manuscript.

Thank you for your kind consideration.

Yours sincerely

Dr. Xinming Wang
State Key Laboratory of Organic Geochemistry
Guangzhou Institute of Geochemistry, Chinese Academy of Sciences
Guangzhou 510640, China.
E-mail: wangxm@gig.ac.cn

Dr. Xiang Ding
State Key Laboratory of Organic Geochemistry
Guangzhou Institute of Geochemistry, Chinese Academy of Sciences
Guangzhou 510640, China.
E-mail: xiangd@gig.ac.cn

I have walked through your responses and changes to the comments by the reviewers. They are generally adequate and have improved the manuscript. I do, however, agree with one of the review comments that it would be better to link the source contribution to the health risks. Specifically, the Section 3.4 currently is still mostly about source contribution; it would be nice to have a bit more discussion on how different emission sources would lead to different health risks.

Reply: Thank you for your suggestion. In the revised manuscript, we add more discussion on how different emission sources would lead to different health risks. And Figure 1-3 were added in supporting information file and the revised manuscript as Figure S15, Figure S16 and Figure S19. (The line numbers here refers to the 'tracking changes' file).

[revised manuscript text omitted]

Figure 1 Source contributions to ILCR in north and south China.

[Figure]

Figure 2 Source contributions to ILCR at urban, sub-urban and remote sites.

[Figure]

Figure 3 Seasonal variations of ILCR source contributions in China.

In addition, a few more minor comments for you to clarify/consider, as below.

1) What was the criteria to choose 10 degree C to separate cold and warm seasons?

Reply: The annual ambient temperature at the 12 sampling sites was 13.9 °C, in the revised manuscript we choose 13.9 °C to divide the one-year data into warm and cold seasons. And Figure 4 was added in supporting information file and the revised manuscript as Figure S10.

[Figure]

Figure 4 Correlation coefficient (r) of PAHs with T (a), SR (b) and BHL (c) at 12 sites in cold and warm season.

*: p<0.05

**: the ambient temperature at BN and SY are all exceed 13.9 °C, there are no cold season in these two sampling sites.**

2) If size segregation is something that distinguishes this study with most of other similar studies in literature, would it be better to reflect it in the title? For example, adding "revealed by size-segregated measurements" at the end of the title?

Reply: Revised as suggested. The title was changed to "Nationwide increase of polycyclic aromatic hydrocarbons in ultrafine particles during winter over China revealed by size-segregated measurements".

3) A few more minor points/typos:

L323 (in track-changed file): suggest to add "was" in front of "partly".

Reply: Revised as suggested. (Line 305)

L428: "dominate" should be "dominating".

Reply: Revised as suggested. (Line 404)

[revised manuscript text omitted]